# Immunotherapy targeting different immune compartments in combination with radiation therapy induces regression of resistant tumors

Nils-Petter Rudqvist [1,8,9,12], Maud Charpentier [1,12], Claire Lhuillier [1,10], Erik Wennerberg [1,11], Sheila Spada [1], Caroline Sheridan[2], Xi Kathy Zhou [3], Tuo Zhang [4], Silvia C. Formenti [1], Jennifer S. Sims[5,6], Alicia Alonso[2] & Sandra Demaria [1,7] ✉

Radiation therapy (RT) increases tumor response to CTLA-4 inhibition (CTLA4i) in mice and in some patients, yet deep responses are rare. To identify rational combinations of immunotherapy to improve responses we use models of triple negative breast cancer highly resistant to immunotherapy in female mice. We find that CTLA4i promotes the expansion of CD4[+] T helper cells, whereas RT enhances T cell clonality and enriches for CD8[+] T cells with an exhausted phenotype. Combination therapy decreases regulatory CD4[+] T cells and increases effector memory, early activation and precursor exhausted CD8[+] T cells. A combined gene signature comprising these three CD8[+] T cell clusters is associated with survival in patients. Here we show that targeting additional immune checkpoints expressed by intratumoral T cells, including PD1, is not effective, whereas CD40 agonist therapy recruits resistant tumors into responding to the combination of RT and CTLA4i, indicating the need to target different immune compartments.

Therapies that activate the immune system to treat cancer have been successful in many tumor types, but in the majority of patients they are insufficient to control the tumor[1]. Focal radiation therapy can enhance immune recognition of the tumor[2], and has been shown to improve responses to CTLA4 inhibition in mice[3,4], and in some patients[5–7]. In pre-clinical studies, radiation enhanced the diversity of the T-cell receptor (TCR) repertoire of intratumoral T cells, while anti-CTLA4 promoted T cell expansion in the tumor and peripheral blood[8,9]. In metastatic lung cancer patients responding to treatment with focal radiation and anti-CTLA4 we have observed a rapid expansion of tumor-specific T cell clones in the peripheral blood. However, T cell proliferation, evidenced by KI67 expression, was seen in both,

[1]Department of Radiation Oncology, Weill Cornell Medicine, New York, NY 10065, USA. [2]Department of Medicine, Weill Cornell Medicine, New York, NY 10065, USA. [3]Division of Biostatistics, Department of Population Health Sciences, Weill Cornell Medicine, New York, NY 10065, USA. [4]Department of Microbiology and Immunology, Weill Cornell Medicine, New York, NY 10065, USA. [5]Human Oncology and Pathogenesis Program, Memorial Sloan Kettering Cancer Center, New York, NY 10065, USA. [6]Immunogenomics and Precision Oncology Platform, Memorial Sloan Kettering Cancer Center, New York, NY 10065, USA. [7]Department of Pathology and Laboratory Medicine, Weill Cornell Medicine, New York, NY 10065, USA. [8]Present address: Department of Thoracic/Head and Neck Medical Oncology, University of Texas MD Anderson, Houston, TX 77030, USA. [9]Present address: Department of Immunology, University of Texas MD Anderson, Houston, TX 77030, USA. [10]Present address: Department of Immuno-Oncology, Sanofi, 94403 Vitry-sur-Seine, France. [11]Present address: Division of Radiotherapy and Imaging, Institute of Cancer Research, London SM2 5NG, UK. [12]These authors contributed equally: Nils-Petter Rudqvist, Maud Charpentier. ✉e-mail: szd3005@med.cornell.edu

responding and progressing patients[6], suggesting a complex interaction between radiation and CTLA4 inhibition.

CTLA4 regulates T cell activation by hindering CD28-mediated signaling on conventional and regulatory T cells[10–12], by enhancing T cell motility[13], and by reducing co-stimulatory molecules expression on antigen-presenting cells, an effect mediated by regulatory T cells (Treg) that express high CTLA4 levels[14,15]. In addition, data in mouse models of genetic CTLA4 deletion indicate that CTLA4 not only limits activation but also constrains the differentiation states of CD4+ T cells. CTLA4 inhibition in tumor-bearing mice resulted in the expansion of effector Th1-like CD4+ T cell subsets similar to the archetypes observed in CTLA4-deficient mice[16], suggesting that reducing the constrains imposed by CTLA4 on the differentiation state of CD4+ T cells may contribute to the therapeutic effects of CTLA4 therapy.

To gain more mechanistic insights about the interaction of radiation therapy and CTLA4 inhibition that may help understand the bases for the success or failure of this combination in patients we used the mouse carcinoma 4T1, which enabled us to interrogate the effects of each therapy alone and in combination. 4T1 is a model of triple-negative breast cancer (TNBC) and is poorly immunogenic, very aggressive and resistant to CTLA4 therapy. Focal radiation therapy is synergistic with CTLA4 inhibition leading to CD8+ T cell-mediated control of the irradiated tumor and its metastases and to occasional cures[3,17,18]. Similar to what we have observed in patients responding to radiation and CTLA4 inhibition, the combined therapy results in the expansion of T cell clones, including 4T1 antigen-specific CD8+ T cell clones[6,9]. However, in prior studies, we used bulk T cell analyses, which did not allow the resolution of the differentiation state of the T cells generated by treatment.

Here we perform longitudinal analyses of the intratumoral TCR repertoire, as well as single cell analysis, and identify separate contributions of each therapy, radiation and CTLA4 inhibition, to the expansion of specific T cell clusters that are associated with improved survival in TNBC and melanoma patients. Whereas CTLA4 inhibition expands CD4 T helper 1 (CD4$_{TH1}$) cells, radiation expands exhausted CD8 T (CD8$_{EX}$) cells. In the tumor of mice treated with both, radiation and CTLA4 inhibition, CD4$_{TH1}$ and CD8$_{EX}$ are present in similar proportion but in addition CD4 regulatory (CD4$_{TREG}$) cells are reduced while CD8 effector memory (CD8$_{EM}$), early activation (CD8$_{EA}$) and precursor exhausted (CD8$_{PEX}$) T cells are expanded compared to control and monotherapy-treated mice. A combined gene signature comprising the three CD8+ T cell clusters expanded in the tumor of mice treated with radiation and CTLA4 inhibition is associated with survival in patients. Despite high expression by intratumoral T cells, inhibition of checkpoint receptors PD-1 and LAG3 expressed by CD8$_{EX}$ T cells does not improve responses to radiation and CTLA4 inhibition, and neither does the targeting of GITR and OX40 that are highly expressed on CD4$_{TREG}$ cells. In contrast, agonistic CD40 therapy leads to deep and complete tumor responses in a majority of 4T1 tumor-bearing mice, an effect that is confirmed in the AT3 tumor model. Overall, these results indicate that targeting multiple checkpoints on T cells may not improve responses in the context of radiation therapy, while targeting a complementary cellular compartment can recruit resistant tumors into responding.

## Results

### Radiation therapy drives an increase in intratumoral TCR repertoire clonality and divergence

We have previously reported the cross-sectional analysis of TCR repertoire changes elicited by radiation therapy (RT) and/or antibody-mediated CTLA4 inhibition (hereafter, CTLA4i) in 4T1 tumors[9]. Although we observed increased clonality in treated mice, and treatment-specific clustering of TCRs that were shared by different mice, the study was limited by the analysis of only *Tcrb* chain at a single time point. To monitor longitudinal changes elicited by therapy in each individual mouse we injected 4T1 cells in both flanks and removed surgically one of the two tumors before start of therapy (pre-tx), while the other tumor was removed at the completion of treatment (post-tx) for deep sequencing of the *Tcra* and *Tcrb* CDR3 regions (Fig. 1a). Post-treatment analyses were performed 10 days after treatment start, when tumor growth curves begin to separate in mice treated with RT versus RT+CTLA4i, (Fig. 1b and Supplementary Fig. 1) as immune-mediated tumor rejection becomes apparent[17,18]. RT was given in three daily doses of 8 Gy similar to the RT regimen that was effective in the clinic at inducing an increase in circulating IFNβ and objective tumor responses with CTLA4i[6]. The induction of IFNβ secretion by 4T1 tumor cells treated with 8GyX3 was confirmed in vitro (Supplementary Fig. 2). Pre-tx TCR repertoire clonality was similar in the different treatment groups. Comparison of TCR repertoires of paired pre- and post-tx tumor samples showed some degree of clonal expansion in treated mice, with the largest increase driven by RT alone or with the addition of CTLA4i (Fig. 1c and Supplementary Fig. 3a). In contrast, no significant changes were seen in untreated mice, indicating that tumor progression by itself did not affect TCR clonality, at least in the time window of observation. We also calculated the cumulative frequency of top 10 clones and all clones with a frequency above 1% which revealed a similar increase as for the clonality assessment (Supplementary Fig. 3b, c). Lastly, we performed the same analysis but after normalizing all TCR repertoires to contain the 500 top clones to account for differences in the number of rearrangements between samples, and obtained similar results as when the entire repertoire was assessed (Supplementary Fig. 3d, e).

Next, we used the VJ-gene-based Jensen-Shannon divergence (JSD$_{VJ}$) to estimate the similarity between paired pre- and post-tx TCR repertoires within each mouse. The intratumoral TCR repertoire of mice treated with RT and RT+CTLA4i showed a significant increase in divergence between pre-tx to post-tx, as compared to untreated or CTLA4i treated mice (Fig. 1d). To determine if the higher divergence observed in RT and RT+CTLA4i treated mice could be explained by the increase in clonality, we calculated the paired rank-based divergence using the top 700 clones in each repertoire (JSD$_{RANK,700}$; this limit allowed for minimal down sampling of each sample) and correlated this to the VJ-gene based JSD of the same clones (JSD$_{VJ,700}$). A correlation coefficient between JSD$_{RANK,700}$ and JSD$_{VJ,700}$ of 1 would indicate that the divergence is completely driven by clonality. The correlation analysis showed that for *Tcra* and *Tcrb*, 55% and 71% of the variation in JSD$_{VJ,700}$ could be explained by variation in JSD$_{RANK,700}$, respectively (Fig. 1e). Altogether, these results show that RT−with or without CTLA4i−significantly increased the clonality of the intratumoral TCR repertoire, and that this was largely responsible for the high divergence observed between pre- and post-tx repertoires.

Next, we investigated the TCR repertoire of CD8+ T cells specific for the tumor epitope AH1 (SPSYVYHQF) that is derived from the gp70 envelope protein of an endogenous retrovirus and is expressed by 4T1 cancer cells[9]. Using CDR3B sequences that define AH1-specific TCRs (see ref. 9 and Methods; full AH1 signature in Supplementary Data 1) we found that the frequency of AH1-reactive T cells was significantly increased after RT used alone or in combination with CTLA4i (Supplementary Fig. 4a). However, clonality of AH1-specific T cells was only significantly increased by RT when used alone (Supplementary Fig. 4b), suggesting that RT fosters clonal expansion of a pre-existing CD8+ T cell response. In contrast, when the TCR repertoire was analyzed after removing the AH1-specific TCRs, each treatment enhanced clonality (Supplementary Fig. 4c). Furthermore, the TCR clonality without AH1-specific TCRs correlated with the overall clonality ($R^2$ = 81%; $p < 10^{-4}$) (Fig. 1f). Thus, the AH1-unrelated TCR repertoires are the main contributors to the changes in clonality observed in mice treated with RT and/or CTLA4i (Fig. 1c), indicating that the majority of the expanded clones found in post-tx tumors are specific for antigens others than AH1.

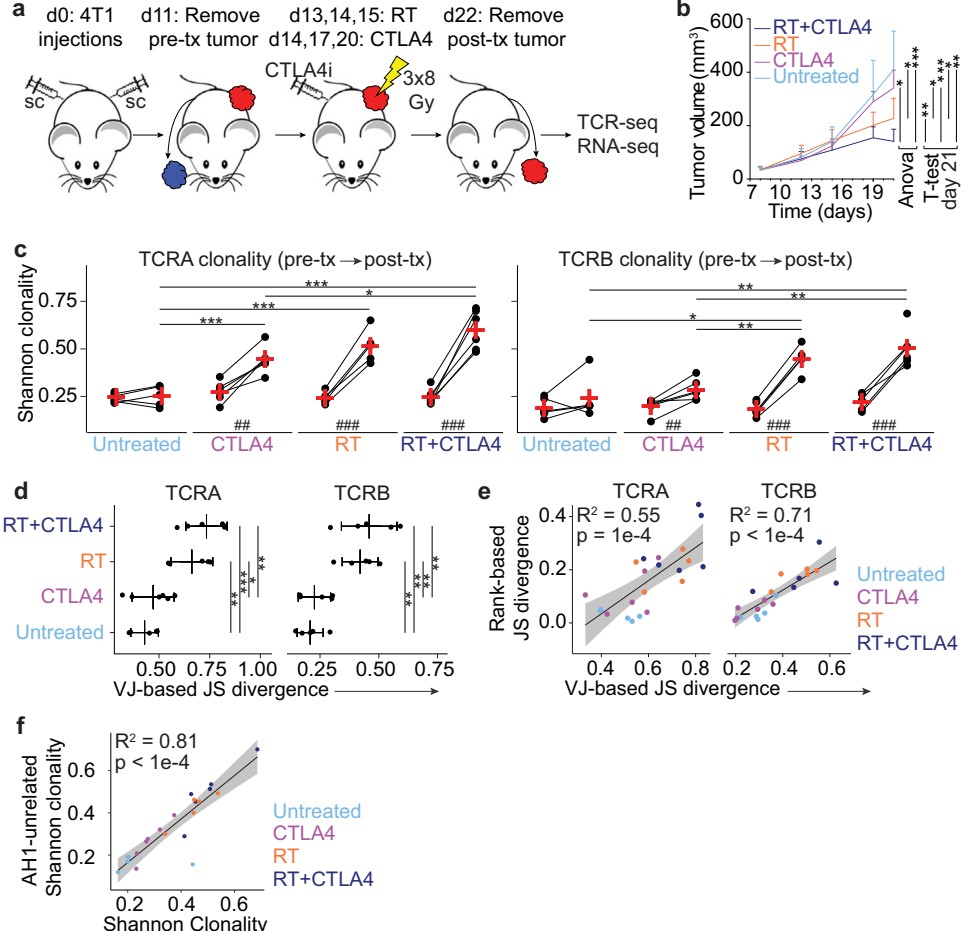

**Fig. 1 | Clonal expansion of T cells in 4T1 tumors post-therapy. a** Experimental schema for collection of pre- and post-treatment (pre-tx and post-tx) 4T1 tumor tissue ($n = 6$ biologically independent mice/group were used for panels **b**–**f**). **b** Tumor growth curves; lines and error bars illustrate mean and standard deviation (error bar only shown in one direction for visualization). Statistical significance in tumor volume growth between groups was determined with 2-way repeated measures ANOVA between day 15–21, and $t$ test at day 21, as indicated in the figure. **c** Shannon clonality of paired pre- and post-tx T cell receptor (TCR) repertoires. Lines indicate paired samples from the same mouse. Red crosses indicate mean value within group/timepoint. *-*** and ##-### indicate p-values for pairwise and paired $t$ tests, respectively. **d** VJ-gene based Jensen-Shannon Divergence (JSD) calculated between paired pre- and post-tx TCR repertoires. *-*** indicate $p$-values pairwise $t$ tests. **e** Linear regression between ranked and VJ-gene based JSD. **f** Linear regression between clonality of unmodified ($x$-axis) and AH1-unrelated ($y$-axis) TCRB repertoires. For all panels: Tukey's and Holm's method for adjusting p-values corrected for multiple comparison was used for the ANOVA and $t$ tests, respectively; *, ** and ***, and #, ##, ###, and ####, indicate $p$-values < 0.05, 0.01, 0.001, and 0.0001, respectively. Panels (**e**) and (**f**): $R^2$ and $p$ indicate linear regression model $R$-squares and $p$-values for the models, respectively, shaded area represents the 95% confidence interval. Source data and exact $p$ values are provided in the Source Data file.

## Radiation therapy-induced T cell increase in tumors is enhanced by CTLA4i

To investigate the nature of the intratumoral T cells associated with the TCR repertoire changes observed, global gene expression was analyzed by RNA sequencing of the same tumors (Fig. 1a). CTLA4i alone did not increase the presence of T cells within the tumors, whereas RT and to a larger extent RT+CTLA4i induced a marked increase in all T cell markers (Fig. 2a), with 46% and 95% of the variation in *Cd3e* expression explained by *Cd4* and *Cd8a* expression, respectively (Fig. 2b). Global differential gene expression analysis comparing treated vs. untreated tumors showed that CTLA4i monotherapy did not induce gene expression changes, whereas RT and RT+CTLA4i treatment changed the transcriptional profile substantially (Fig. 2c). Transcriptional profiles of tumors treated with RT and RT+CTLA4i were similar, although the combination therapy yielded the most distinct gene expression profiles compared to untreated tumors (Fig. 2d). Principal component analysis of the gene expression profiles also revealed clustering of post-tx tumors based on whether they received RT or not (Fig. 2e).

Canonical pathway and upstream regulator analysis were used to interrogate the biological activity of the differentially expressed genes. Almost all of the canonical pathways with a significant z-score were activated (i.e., $z > 2$), and a majority of these were immune related, e.g., activation of Th1 pathway (z-scores: 2.2 for RT; 3.4 for RT+CTLA4i), iCOS-iCOSL signaling in T helper cells (2.8 for RT+CTLA4i), Cytotoxic T Cell-mediated Apoptosis of Target Cells (z-scores: 2.0 for RT; 2.4 for RT+CTLA4i), and CD28 Signaling in T Helper Cells (z-score: 2.2 for RT+CTLA4i) (Fig. 2f). Analysis of upstream transcriptional regulators of the observed differentially expressed genes predicted upstream activation of many immune related proteins, in particular following RT+CTLA4i, including IFNG (z-scores: 4.9 for RT; 8.3 for RT+CTLA4i), TNF (z-scores: 3.6 for RT; 7.2 for RT+CTLA4i), IFNA (z-scores: 3.7 for RT; 6.1 for RT+CTLA4i), and STING (z-scores: 3.1 for RT; 4.8 for RT+CTLA4i), consistent with the activation of an anti-tumor immune response (Fig. 2g). Consistently, immune suppressive proteins SOCS1 (z-scores: −3.4 for RT; −4.7 for RT+CTLA4i) and IL10RA (z-scores: −6.0 for RT+CTLA4i) were predicted to be inhibited. Altogether, these data

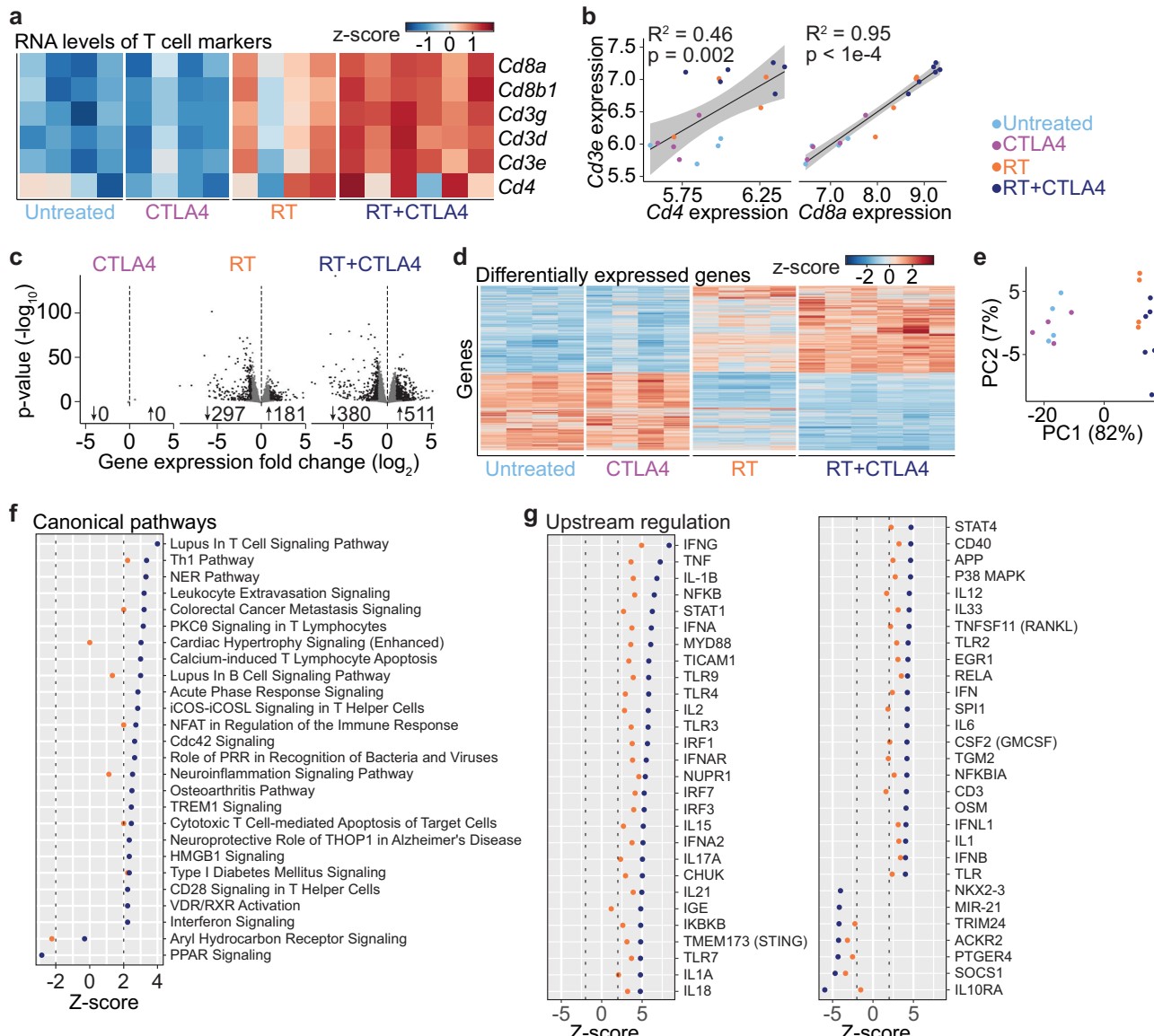

**Fig. 2 | Increased intratumoral T cells following RT and RT + CTLA4i treatment.**
Bulk RNA-sequencing was performed on whole 4T1 tumors ($n = 4$ biologically
independent mice for control, CTLA4, and RT groups, $n = 6$ biologically indepen-
dent mice for RT + CTLA4). **a** Gene expression (scaled DEseq2 counts) of a selection
of canonical T cell markers in post-treatment tumors. Each column represents an
individual mouse. **b** Linear regression between *Cd3e* and *Cd4* or *Cd8* expression in
post-treatment tumors. $R^2$ and *p* indicate linear regression model *R*-squares and *p*-
values for the models, respectively, shaded area represents the 95% confidence
interval. Each dot represents an individual mouse. **c** Volcano plots of differentially
expressed genes. CTLA4i, radiation therapy (RT), and RT+CTLA4i treated tumors
were compared with untreated tumors (using DEseq2 on GEX counts data). Genes

with adjusted *p*-value < 0.01 and |fold change| > 2 were considered differentially
expressed (number of genes shown with up- or down-arrows indicating increased
or decreased expression, respectively). **d** Heatmap showing normalized and scaled
expression of all differentially expressed genes (for any comparison). Each column
represents an individual mouse. **e** Principal component (PC) analysis of the global
gene expression profiles. The proportion of variation explained by PC dimensions 1
and 2 are shown in parentheses. Each dot represents an individual mouse.
**f**, **g** Ingenuity Pathway Analysis Canonical Pathway and Upstream Regulation ana-
lysis. Z-scores indicate predicted activation (>2) or inhibition (<−2) of pathways and
upstream regulators. Source data are provided as a Source Data file.

demonstrate the ability of RT+CTLA4i to convert an immunologically
cold tumor into a hot tumor.

## Distinct CD4⁺ and CD8⁺ T cell differentiation states are induced by radiation therapy and CTLA4i

Results of bulk RNA-seq analyses indicated that the direction of
changes in immune gene expression was overall similar in tumors of
RT and RT+CTLA4i treated mice. However, some pathways like the
ICOS/ICOSL pathway, which has been implicated in responses to
CTLA4i[19], were only upregulated in RT+CTLA4i treated mice, sug-
gesting an effect of CTLA4i in regulating T cell functional

differentiation[16], in addition to expanding T cells primed by RT. To gain
a better understanding of the T cell response shaped by RT and
CTLA4i, T cells infiltrating 4T1 tumors were sorted and analyzed by
single-cell sequencing (Supplementary Fig. 5). The data obtained from
each of the four treatment groups (untreated, RT, CTLA4i, and RT
+CTLA4i) were integrated using Seurat[20]. To reduce bias due to the
number of cells, 1920 cells, close to the lowest number of cells from
any group, were randomly selected from each group prior to analysis,
resulting in a total of 7680 T cells being used for downstream analysis.
Unbiased clustering of the T cells based on gene expression revealed 17
distinct clusters (hereinafter referred to as C0-C16), visualized in 2D

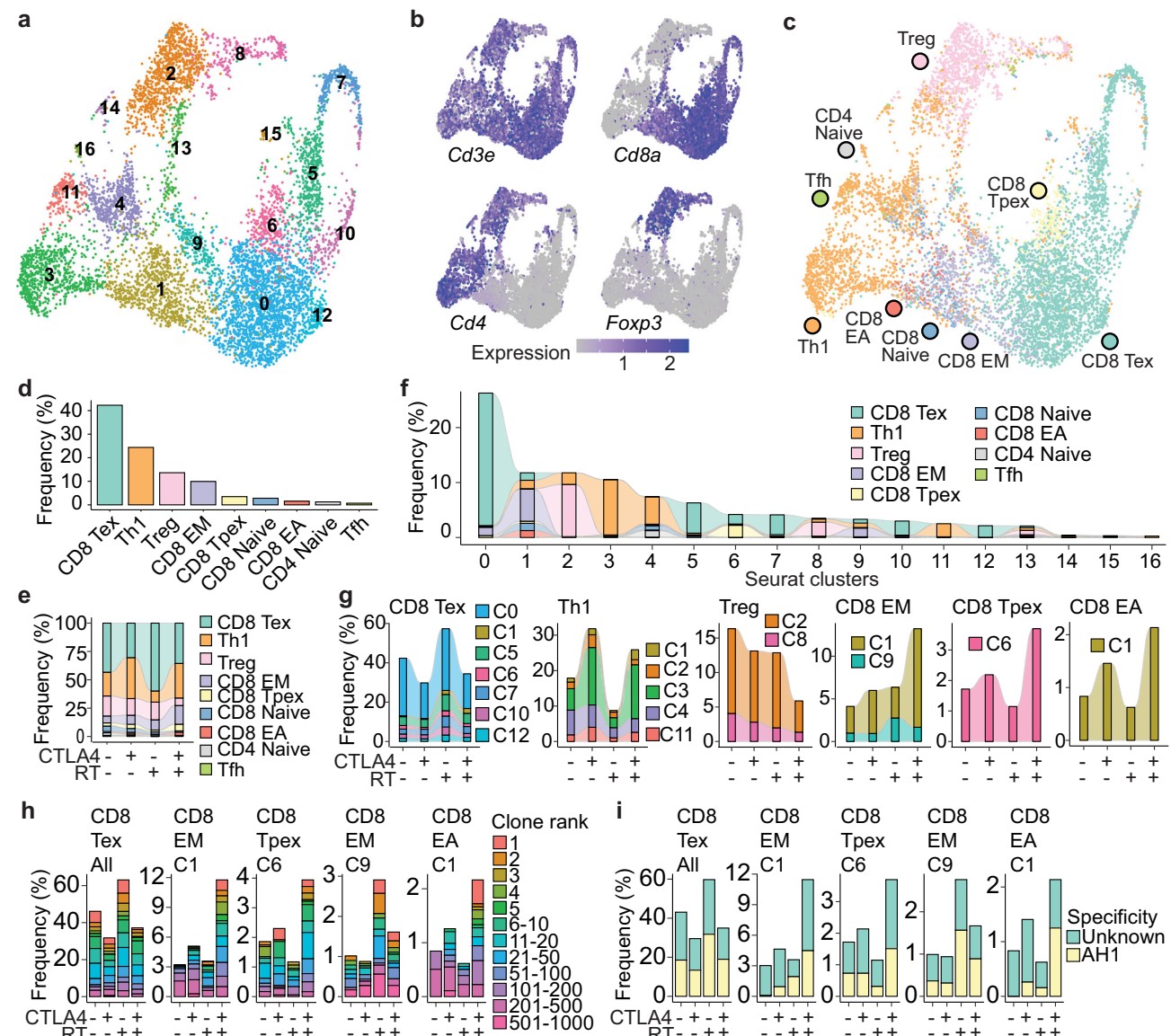

**Fig. 3 | Distinct CD4+ and CD8+ T cell differentiation states are induced by RT and CTLA4i.** Single-cell RNA and TCR sequencing were performed on Cd3+Cd4+Cd8- and Cd3+Cd4-Cd8+ cells (T cells) infiltrating post-tx 4T1 tumors (tumors from 5 biologically independent mice per treatment group were pooled for scRNAseq analysis). **a** T cells were divided into 17 clusters (0–16; indicated by colors) and visualized in 2D using the UMAP dimensionality reduction algorithm. **b** Relative gene expression of canonical T cell markers. **c** Projection on UMAP and (**d**) overall frequency, and (**e**) treatment group specific frequency of T cells mapped to different functional states based on ProjecTILs annotation[22]. Tex: terminally-exhausted; Th1; T helper type 1; Treg: T regulatory; EM: effector-memory; Tpex: precursor-exhausted; EA: early-activation; Tfh: T follicular helper. **f** Overall and (**g**) treatment group-specific mapping of T cells for a selection of ProjecTILs functional state to clusters as defined using the Seurat pipeline. Treatment group-specific distribution of (**h**) T cell clones and (**i**) AH1-specific T cells among a selection of ProjecTILs functional states. Source data are provided as Supplementary Data 2.

following dimensionality reduction using the Uniform Manifold Approximation and Projection (UMAP) algorithm (Fig. 3a, heatmap in Supplementary Fig. 6)[21]. Clusters clearly segregated as CD8+ or CD4+, the latter containing suppressive Foxp3+ CD4_TREG cells and conventional CD4+ T cells (Fig. 3b). To deconvolute the major transcriptional states of the T cells, the ProjecTILs computational method was used applying the "tumor-infiltrating T lymphocytes (TIL) atlas" as a reference dataset[22] (Fig. 3c, Supplementary Fig. 7). This analysis revealed that more than 90% of all T cells were associated with one of four major transcriptional states: CD8_EX (42%), Th1-like CD4+ T cells (CD4_TH1, 24%), CD4_TREG, (14%), and CD8_EM (10%) (Fig. 3d).

Next, we assessed if treatment altered the relative representation of each ProjecTILs-defined T cell subset using statistical significance thresholds of $p < 0.05$ and odds ratio (OR) > 1.5 or <−1.5 (Fig. 3e,

statistical analysis data in Supplementary Table 1). In untreated tumors, CD8_EX (43.3%), CD4_TH1 (21.0%), and CD4_TREG (17.7%) cells were the most prominent, followed by CD8_EM (6.1%), CD8 naïve (4.8%), precursor exhausted CD8+ T cells (CD8_PEX, 2.8%), CD4 naïve (2.3%), early activation CD8+ T cells (CD8_EA, 1.1%), and lastly CD4+ T follicular helper cells (T_FH, 1.0%). CTLA4i enhanced the proportion of CD4_TH1 (from 21% to 36.1%) at the expense of CD8_EX cells (from 43.3% to 30.4%). In contrast to CTLA4i, RT enriched CD8_EX (from 43.3% to 59.8%) and CD8_EM (from 6.1% to 9.1%) cells, while reducing CD4_TH1 (from 21.0% to 9.8%) and CD8_PEX (from 2.8% to 1.7%) cells. The proportion of CD4_TREG was not significantly different in RT (15.5%) or CTLA4i (14.7%) as compared to untreated (17.7%) tumors. In the RT +CTLA4i treated tumors, T cell subsets reflected the influence of both therapies: CD8_EX cells decreased but remained the dominant

phenotype (from 43.3% to 35.5%) and CD4$_{TH1}$ cells significantly increased (from 21.0% to 30.5%). In addition, RT+CTLA4i significantly reduced CD4$_{TREG}$ (from 17.7% to 6.6%) and enriched for CD8$_{EM}$ (from 6.1% to 16.5%), CD8$_{EA}$ (from 1.1% to 2.8%), and CD8$_{PEX}$ (from 2.8% to 5.9%) cells. Naïve CD4 and CD8 T cells were reduced in proportion in each treated group. Altogether, these data suggest an important qualitative difference in the intratumoral T cell populations that are increased by RT versus RT+ CTLA4i. Whereas CD8$_{EX}$ are the dominant population in RT-treated tumors, RT + CTLA4i selectively expand CD8$_{EA}$ and CD8$_{PEX}$ T cells, while also driving a larger expansion of CD8$_{EM}$ T cells as compared to RT alone. In the CD4 compartment CD4$_{TH1}$ cells are increased and CD4$_{TREG}$ reduced (Fig. 3e).

Next, we asked if T cells belonging to the same ProjecTILs-defined subset were found within different Seurat clusters. If true, we hypothesized that T cells belonging to a defined T cell state may acquire different transcriptional profiles in the TME, possibly indicating a transition in the functional state. To focus this analysis on major subsets, we included only ProjecTILs-defined subsets that populated >1% of T cells in ≥ 1 Seurat cluster. Then, for each included ProjecTILs-defined subset, a Seurat cluster was included if (i) > 1% of cells in it was assigned to the specific ProjecTILs subset and if (ii) > 10% of cells in the ProjecTILs subset was allocated to the specific Seurat cluster. Based on these inclusion criteria, CD8$_{EX}$ cells were found in C0, C1, C5, C6, C7, C10, and C12; CD4$_{TH1}$ cells in C1, C2, C3, C4, and C11; CD4$_{TREG}$ cells in C2 and C8, CD8$_{EM}$ cells in C1 and C9; CD8$_{PEX}$ cells and CD8$_{EA}$ cells were present in a single cluster, C6 and C1, respectively (Fig. 3f). Naïve T cells were not included in downstream analysis. Only two Seurat clusters were associated with multiple ProjecTILs-defined subsets: C1, which included CD8$_{EX}$, CD4$_{TH1}$, and CD8$_{EM}$ cells, and C2 which included CD4$_{TH1}$ and CD4$_{TREG}$ cells.

To characterize the ProjecTILs-defined subsets within each Seurat cluster, a differential gene expression analysis was performed (Supplementary Data 2, Supplementary Data 3). We then used the literature to interpret the function of each T cell subset.

Among CD8$_{EX}$ cells (Supplementary Fig. 8; Supplementary Data 2; Supplementary Data 3), 57% were found in C0, a cluster defined by differential expression of *Gzma*, *Gzmb*, and *Nkg7* that encode effector cytotoxic molecules[23], *Pdcd1* that encodes the inhibitory receptor PD1, and *Klra5* which is associated with terminally differentiated effectors[24]. These cells also differentially express *Lgals3* that encodes Galectin 3, which in the tumor microenvironment has been shown attenuate T cell infiltration by capturing *Ifng*[25]. CD8$_{EX}$ cells in C5 (13%), C7 (3.6%), and C10 (6.1%) were defined by expression of genes associated with cell cycle and/or proliferation, (*Mcm3*, *Mcm5*, *Mcm6* in C5, *Ccna2*, *Ccnb2*, and *Mki67* in C7, and *Ccnb2* and *Cenpa* in C10). Similarly to C0, CD8$_{EX}$ cells in C12 (5%) differentially expressed *Gzma* and *Gzmb*, but contrasted by high expression of other granzymes (in particular *Gzmd*, *Gzme*, and *Gzmg*) with less defined substrates[26]. *Prf1*, encoding the cytolytic molecule perforin which acts to perforate the membrane of target cells, was also highly expressed in this subset, suggesting a more cytotoxic profile of CD8$_{EX}$ cells in C12. Lastly, CD8$_{EX}$ cells in C12 exhibited high expression of *Irf8* and *Spp1* that in 4T1 tumors have been associated with balancing generation of antigen-specific CD8$^+$ T cells and tumor rejection[27]. CD8$_{EX}$ cells in C1 and C6 had no genes passing the differential expression significance threshold; T$_{EM}$ and T$_{EA}$ CD8 T cells were present in C1, while C6 contained CD8$_{PEX}$ cells, suggesting that CD8$_{EX}$ cells in these clusters may represent transition states.

Among the CD4$_{TH1}$ cells (Supplementary Fig. 9; Supplementary Data 2; Supplementary Data 3), 41% was found in C3 and was defined by differential expression of the transcriptional regulator *Bhlhe40*, which acts to promote Th1 responses by induction of IFN-γ and inhibition of IL-10[28], *Csf2*, encoding the pro-inflammatory cytokine GM-CSF, and *Lta*, encoding lymphotoxin-α. *Cd40lg* and *Tnfsf11* (*Rankl*) were also expressed by CD4$_{TH1}$ cells in C3, suggesting a role in improving

functionality and survival of dendritic cells within the tumor[29,30]. Lastly, *Il13*, and *Ccl1*, were differentially expressed in a portion of the cells in this cluster, suggesting a polyfunctional phenotype of some of these cells[31]. CD4$_{TH1}$ cells in C4 (20.5%) were defined by expression of *Socs3* and *Tcf7*, which encode proteins that suppress IFNγ production in favor of Th2 cytokines[32]. However, they did not express cytokines and had high levels of *Dusp10* (*Mkp5*) which has been implicated in reducing effector T-cell cytokine expression[33]. Thus, CD4$_{TH1}$ cells in C4 may represent a somewhat dysfunctional CD4$^+$ T cell subset that may be more Th2- than Th1-like. Similarly, CD4$_{TH1}$ cells in C11 were defined by differential expression of the master Th2 transcription factor *Gata3* and cytokines *Il4* and *Il5*, *Tnfsf8* (*Cd30l*), *Cd40lg*, *Cdkn1a* (*p21*), and *Dusp10*, suggesting these are Th2- and not Th1-like T cells. Interestingly, CD4$_{TH1}$ cells in C2 express CD4$_{TREG}$ markers such as *Foxp3*, *Izumo1r* (*Juno*), and *Ikzf2* (*Helios*), suggesting these cells are in fact CD4$_{TREG}$ and not Th1-like T cells.

Among CD4$_{TREG}$ cells (Supplementary Fig. 10; Supplementary Data 2; Supplementary Data 3), 70% were found in C2 and defined by differential expression of *Ctla4*, *Foxp3*, *Il2ra* (*Cd25*), *Tnfrsf18* (*Gitr*), *Tnfrsf4* (*Ox40*), and *Tnfrsf9* (*Cd137*). They also expressed *Ikzf2* (*Helios*), which has been associated with CD4$_{TREG}$ phenotype and *Foxp3* expression stability[34]. CD4$_{TREG}$ cells in C8 (19%) differentially expressed *Mki67*, suggesting these cells are actively proliferating. This is also suggested by the UMAP plot showing that C8 is adjacent to the other clusters associated with cell proliferation and cell cycling (Fig. 3a).

CD8$_{EM}$ cells (Supplementary Fig. 11; Supplementary Data 2; Supplementary Data 3) were found mostly in C1 (59%) and were defined by differential expression of *Ccl5*, *Gzmk*, *Ly6c1*, and *Ly6c2*, supporting the classification of these cells[35]. In C9, CD8$_{EM}$ cells (16%) were characterized by the expression of genes involved in the interferon type 1 pathway and anti-viral activity (e.g., *Isg15*, *Isg20*, and *Mx1*)[36].

CD8$_{PEX}$ (Supplementary Fig. 12; Supplementary Data 2; Supplementary Data 3) were found in C6 and their gene expression profile was consistent with the gene signature for T$_{PEX}$ defined by ProjecTILs. Likewise, CD8$_{EA}$ (Supplementary Fig. 13; Supplementary Data 2; Supplementary Data 3) were found in C1 and confirmed to have the expected ProjecTILs expression profile[22].

To better understand the effects of treatment on the transcriptional states within ProjecTILs-defined subsets, we analyzed the direction of change of Seurat cluster (Fig. 3g, statistical analysis data in Supplementary Table 2). CD4$_{TH1}$ cell expansion driven by CTLA4i (Fig. 3e) was mainly explained by an enrichment in C3 (from 6.0% in control to 16.2% in CTLA4i and 15.2% in RT+CTLA4i). In contrast, RT decreased CD4$_{TH1}$ cells in clusters C3 (from 6.0% in control to 2.8% in RT) and C4 (from 7.0% in control to 2.9% in RT) and the decrease in C4 was not rescued by CTLA4i in the RT+CTLA4i group.

CD8$_{EX}$ cell contraction driven by CTLA4i was explained by a decrease in C0 (from 29.3% in control to 18% in CTLA4i and 17.7% in RT+CTLA4i). Interestingly, the expansion of CD8$_{EX}$ in RT-treated tumors was largely driven by clusters C5 (from 4.5% to 8.3%), C7 (from 2.5% to 5.6%), and C10 (from 2.2% to 4.1%), indicating that these CD8$_{EX}$ cells are proliferating and likely came into the tumor after RT[37]. However, these clusters were not expanded in RT+CTLA4i-treated tumors. A small but significant expansion of T$_{EX}$ in C12 (from 1.4% in control to 3.3% in RT) was also seen only in RT-treated tumors.

In contrast to the effect on T$_{EX}$, RT led to a contraction of proliferating CD4$_{TREG}$ in C8 (from 4.1% in control to 2% in RT and 1.4% in RT+CTLA4i). A significant decrease of CD4$_{TREG}$ in C2 was seen only in RT+CTLA4i treated tumors (from 12.3% in control to 4.5% in RT+CTLA4i), explaining the overall decrease in CD4$_{TREG}$ observed only with the combination treatment.

Among CD8$_{EM}$ T cells, C9 was significantly expanded by RT alone (from 1.0% to 2.8%), which is in line with increased interferon type 1 signaling in irradiated tumors[38]. Finally, in RT+CTLA4i treated tumors there was a variable but significant expansion of C1 in several

 

ProjecTILs-defined subsets, namely, $CD8_{EA}$, $CD8_{EM}$, $CD8_{EX}$, and $CD4_{TH1}$ T cells, suggesting that cells in this cluster may be defined by shared activation signatures.

In summary, CTLA4i and RT have distinct effects on intratumoral T cells with CTLA4i driving $CD4_{TH1}$ T cells and RT bringing into the tumor CD8 T cells that are proliferative but exhausted. When combined, RT and CTLA4i reprogrammed the T cell landscape by leading to the expansion of $CD8_{PEX}$, $CD8_{EA}$, $CD8_{EM}$, in addition to expanding activated $CD4_{TH1}$ cells while reducing dysfunctional and regulatory CD4 T cells.

## CD8[+] T cell functional subsets expanded in the TME of RTplus CTLA4i treated mice contain tumor antigen-specific T cells

In the single-cell sequencing experiment, we determined the TCR clonotype of >90% of the T cells in each treatment group and used feature barcoding to identify CD8[+] T cells reactive against the AH1 epitope. To assess the clonality of the T cells within each of the major ProjecTILs-defined subsets and their clusters all TCR clones were ranked (combining CD8[+] and CD4[+] T cells) based on their frequency within each treatment group and then their association with the different T cell subsets was determined. While a majority of clonally expanded T cells in untreated tumors was associated with the $CD8_{EX}$ phenotype, in RT+CTLA4i treated tumors the top clones shifted towards a more functional phenotype and were found in the $CD8_{EM}$/C1 and $CD8_{EM}$/C9, $CD8_{PEX}$, and $CD8_{EA}$ clusters (Fig. 3h). CTLA4i as monotherapy induced clonal expansion of the $CD8_{PEX}$ and $CD8_{EA}$ subsets, whereas RT induced clonal expansion of $CD8_{EX}$ cells, and of the $CD8_{EM}$/C9 T cells. Next, we analyzed the AH1 antigen-specific CD8[+] T cells and found that they were present in multiple transcriptional states in the TME (Fig. 3i). Interestingly, AH1-specific clones were not present within the $CD8_{EM}$/C1 and $CD8_{EA}$ subsets in untreated tumors but represented 40-60% of AH1-specific clones in RT+CTLA4i, consistent with the activation of tumor antigen-specific CD8 T cells by the combination therapy. Clonality was low in CD4 as compared to CD8 T cells in both $CD4_{TH1}$ and $CD4_{TREG}$ subsets, with a slight increase in clonality driven by CTLA4i in $CD4_{TH1}$ but little change in $CD4_{TREG}$ cells regardless of treatment (Supplementary Fig. 14).

To confirm that the CD8[+] T cell subsets emerging in RT+CTLA4i treated mice were detectable consistently in the tumor of individual mice, cluster-specific gene signatures were constructed and used to interrogate the bulk RNA sequencing data by calculating log2 fold-change weighted enrichment scores based on each gene signature. This analysis confirmed the marked increase of all CD8 T cell subsets in RT+CTLA4i treated vs. untreated tumors, with a more modest increase in RT monotherapy-treated tumors (Fig. 4a). Among all CD8 clusters evaluated $CD8_{PEX}$ were selectively increased by the combination treatment while RT had almost no effect. CTLA4i by itself did not induce changes in the T cell signatures in the tumor suggesting that CTLA4i mostly impacts the polarity of the T cell response and not tumor infiltration.

## T-cell differentiation clusters enriched in the tumor of mice treated with radiation therapy and CTLA4i are associated with survival in patients

To assess if the T cell subsets that characterized the T cell landscape in RT+CTLA4i treated 4T1 tumors might be relevant to patients, we tested if the gene signatures that defined T cell clusters $CD8_{EM}$/C1, $CD8_{EA}$, and $CD8_{PEX}$ were associated with outcome in patients with TNBC using the METABRIC gene expression dataset[39]. Enrichment scores were calculated for each patient and their association with survival was assessed using a cox proportional hazard regression model. While there was an association with survival for $CD8_{EM}$/C1 and $CD8_{PEX}$ using an univariate analysis (Fig. 4b), in a more robust multivariate analysis that also considered age at diagnosis, CD3E expression (as a marker for

T cell infiltration that has been associated with survival[40]), menopausal state, and Nottingham Prognostic Index (NPI; which reflects the size, number of involved lymph nodes, and grade of the tumor), no association was found (Fig. 4c). We next decided to test whether all three cell subsets were necessary to predict survival and constructed a combined gene signature that incorporated the gene signatures of all three clusters that were uniquely enriched in tumors after RT+CTLA4i therapy ($CD8_{EM}$/C1, $CD8_{EA}$, and $CD8_{PEX}$). This combined gene signature showed an association with survival in the multivariate analysis (Fig. 4d), suggesting that all three clusters may be necessary to mount an effective immune response. Lastly, we assessed the association with survival for combined signature using a log-rank statistical model. Enrichment scores were calculated for each patient, and patients were classified as having a "high" or "low" score, based on median value. This analysis showed that patients with a high enrichment score for the combined signature had a better prognosis (Fig. 4e). We performed a similar survival analysis in patients with melanoma using the Cancer Genome Atlas skin cutaneous melanoma dataset (TCGA SKCM). We found an association of the combined gene signature with survival when using a cox proportional hazard regression model that corrected for age, AJCC tumor stage, CD3E expression in patients with melanoma ($p < 0.01$) (Fig. 4f). Similarly, results obtained using a log-rank statistical model showed that patients with a high enrichment score for the combined signature had a better prognosis (Fig. 4g). Altogether, these results suggest that the CD8[+] T cells phenotypes that were uniquely expanded in the 4T1 tumor of mice treated with RT+CTLA4i share similarity with T cell functional states present within human tumors in patients who develop spontaneous anti-tumor immune responses.

## Among multiple actionable targets only agonistic CD40 antibody enhanced tumor rejection in mice treated with radiation therapy and CTLA4i

Treatment with RT+CTLA4i improves 4T1 tumor control above what achieved with RT alone but complete tumor regression occurs in only a minority of the mice[3,17,18]. Thus, we reasoned that despite the expansion of T cell phenotypes associated with response to ICI such as $CD8_{PEX}$, $CD8_{EA}$, and $CD8_{EM}$[41], the large $CD8_{EX}$ subset, which expresses multiple inhibitory receptors based on single cell data, was not reinvigorated by CTLA4i. To assess the expression of potentially actionable inhibitory receptors on CD8 T cells infiltrating 4T1 tumors, flow cytometry was performed at day 22 post-treatment, corresponding to the same timeline as the single cell experiment (Supplementary Fig. 5). This analysis revealed that the density of CD8 T cells was markedly increased by RT+CTLA4i (Fig. 5a), consistent with the bulk RNA sequencing data (Fig. 2a), and most of the CD8 T cells were antigen-experienced (Fig. 5b and Supplementary Fig. 15a). A small subset of antigen-experienced CD8 T cells expressed early activation markers (Fig. 5c), and 54% were PD1+ (Supplementary Fig. 15b). Among PD1+ CD8 T cells, based on the level of expression of PD1 and the co-expression of other inhibitory receptors we identified four main clusters (Fig. 5d, e) using the opt-sne dimensionality-reduction and Flow-SOM clustering algorithms on the cloud-based online OMIQ platform[42]. Cluster 1 expressed the highest levels of PD1, TIM3, and TIGIT suggesting that it likely represents the most exhausted T cells (Supplementary Fig. 15c, d). Clusters 2 and 3 expressed progressively decreasing levels of PD1, TIM3 and TIGIT. Cluster 4, the largest cluster, had the lowest expression of PD1 and was negative for TIM3 but expressed the highest levels of LAG3 among all clusters. The percentage of proliferating cells was similar across all clusters (Supplementary Fig. 15d).

In the CD4 T cell compartment, RT+CTLA4i did not significantly increase total T cell density, but the ratio of conventional to regulatory T cells was significantly increased (Fig. 5f, g), whereas CD69 expression remained low in all treatment groups (Fig. 5h). Expression of OX40 was

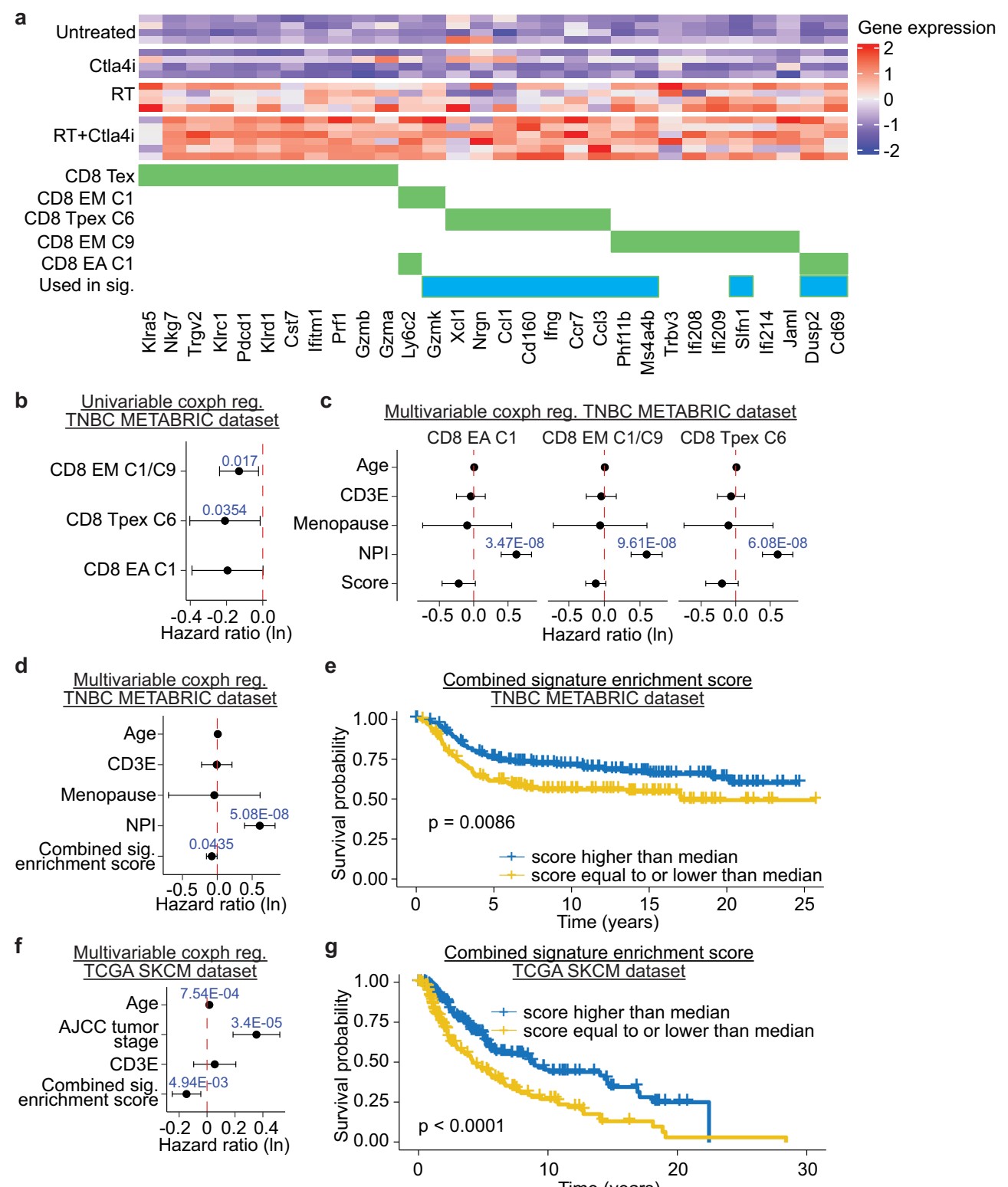

significantly reduced on conventional CD4 T cells in mice treated with RT+CTLA4i compared to control, while the glucocorticoid-induced tumor necrosis factor receptor family related protein (GITR) was expressed by half of the conventional CD4+ T cells without significant difference across all treatment groups (Fig. 5i, d j). OX40 and GITR were also expressed by a majority of regulatory T cells, with a significant increase in the percentage of positive cells driven by RT (Fig. 5k, l, and Supplementary Fig. 16a, b).

Based on these results showing that 36.9% of PD1+ CD8 T cells (cluster 4) expressed the highest levels of LAG3, but were negative for TIM3 and had low levels of TIGIT, while the most exhausted CD8 T cells in cluster 1 only represented 11% of the PD1+ CD8 T cells, we reasoned that targeting PD1 and LAG3 could enhance tumor rejection. Treatment with antibodies blocking PD1 was started after RT and the first dose of CTLA4i, to avoid the detrimental effects of PD1 blockade before T cell priming[43] (Fig. 6a). A similar

**Fig. 4 | Enrichment of functional states has prognostic value in cancer patients.**
**a** Heatmap showing bulk RNA sequencing gene expression levels of differentially expressed genes for different T cells subsets in post-tx 4T1 tumors. Each raw represents an individual mouse ($n = 4$ biologically independent mice for control, CTLA4, and RT groups, $n = 6$ biologically independent mice for RT + CTLA4). Ctla4i and RT indicate CTLA-4 checkpoint blockade and radiation therapy, respectively. Green boxes below heatmap indicate that genes were statistically significantly differentially expressed for a specific functional state in the single-cell sequencing data. Blue boxes indicate genes used in the combined gene signature in (**d**–**g**). **b** Univariate and (**c**) multivariate cox proportional hazard regression models were used to assess the association between gene enrichment score and survival in patients with triple-negative breast cancer (TNBC) using the METABRIC dataset ($n = 299$ biologically independent samples). **d** Multivariate cox proportional hazard regression model and

(**e**) Kaplan–Meier curves of survival of patients with TNBC from the METABRIC dataset by enrichment of a combined gene signature that incorporated differentially expressed genes from CD8 EA/C1, CD8 EM/C1, and CD8 Tpex/C6. Difference in survival between the two groups was assessed using the log rank test. **f** Multivariate cox proportional hazard regression model and (**g**) Kaplan–Meier curves of survival of patients with skin cutaneous melanoma from TCGA SKCM dataset ($n = 417$ biologically independent samples) and enrichment of a combined gene signature that incorporated differentially expressed genes from CD8 EA/C1, CD8 EM/C1, and CD8 Tpex/C6, $p$-value was based on the log rank test. NPI Nottingham Prognostic Index, CD3E expression of CD3E in tumors, AJCC American Joint Committee on Cancer. Bars represent the 95% confidence interval for the hazard ratio (**b**–**d**, **f**). Source data are provided as a Source Data file.

schedule was used for anti-LAG3[44], and administration of both antibodies was continued until tumor progression, as determined by increasing tumor volume recorded on three consecutive measurements for each animal in the group. Somewhat surprisingly, neither anti-PD1 nor anti-LAG3 improved tumor control achieved by RT+CTLA4i (Fig. 6b, c, Supplementary Fig. 17a).

Expression of GITR by CD4 Tconv and Treg cells provided a potential target for expanding effector CD4 T cell while at the same time countering Treg-mediated suppression[45,46]. To target GITR we used the agonistic DTA-1 antibody, previously shown to reduce intratumoral Treg cells and synergize with CTLA4i in inducing the regression of established tumors[46,47]. However, no improvement in tumor control was observed (Fig. 6b, c, Supplementary Fig. 17a). Next, given the ability of PD1 blockade to work in concert with anti-GITR therapy to overcome resistance[45], we tested if the addition of both, anti-PD1 and anti-GITR, could improve responses to RT+CTLA4i, but this strategy did not improve responses of 4T1 tumors.

Expression of OX40 was high on Treg in all treatment groups, consistent with prior reports[48], with the highest levels in the tumors treated with radiation alone (Fig. 5k). OX40 was also expressed by conventional CD4 T cells, although positive cells were reduced in the tumor of mice treated with RT+CTLA4i (Fig. 5i). Agonistic OX40 antibodies have been shown to enhance tumor rejection by stimulating effector T cells while inhibiting the suppressive function of intratumoral Tregs[48,49]. Thus, we reasoned that targeting OX40 with the agonistic antibody OX-86 could improve tumor rejection in mice treated with RT+CTLA4i, but this combination failed to show any benefits (Fig. 6b, c, Supplementary Fig. 17a).

*CD40lg* is one of the genes that define CD4$_{THI}$ in ProjecTIL[22] and it was expressed by the subset identified as CD4$_{THI}$ in our single cell analysis (Supplementary Fig. 9), which was increasingly represented in the tumor of mice treated with CTLA4i, alone or in combination with RT (Fig. 3e, g). However, the percentage of conventional CD4 T cells that were positive for CD40LG by flow cytometry remained low, ~4% even in mice treated with CTLA4i (Supplementary Fig. 16c, d). Thus, we hypothesized that the stimulation of this pathway provided by CD4$^+$ T cells within the tumor was not sufficient, and tested the benefit of an agonistic CD40 antibody[50]. Consistent with this hypothesis, tumor control was significantly improved in mice treated with RT+CTLA4i+anti-CD40 as compared to RT+CTLA4i, with the majority of mice showing tumor regression that was durable in some mice (Fig. 6b, c and Supplementary Fig. 17a). To determine if all three therapies were required to achieve this result, we next compared the response to anti-CD40 + CTLA4i without RT, RT+anti-CD40, and RT+CTLA4i+anti-CD40. Only the triple combination treatment resulted in significant tumor regression (Fig. 6d–f and Supplementary Fig. 17b). Mice that achieved durable complete responses were able to reject a rechallenge with 4T1 cells 90 days later, indicating the presence of a protective immunological memory (Supplementary Fig. 17c).

## Agonistic CD40 antibody reprograms the myeloid compartment in the tumor and draining lymph node

The marked improvement in tumor rejection achieved with anti-CD40 suggested a critical role for increased antigen cross-presentation and T-cell activation in this response. However, in some tumor models CD40 agonism was shown to activate the tumoricidal activity of macrophages that eliminated tumor cells independently from T cells[51]. Thus, we first asked if tumor control was mediated by CD8 T cells in mice treated with RT+CTLA4i+anti-CD40. CD8 T cell depletion abrogated tumor control and decreased mice survival (Supplementary Fig. 18), indicating that CD8 T cells are required for the response to RT+CTLA4i+anti-CD40. To determine the effect of anti-CD40 on cross-presenting conventional dendritic cells type 1 (cDC1) we performed RNAseq of the 4T1 tumors and interrogated the data for the presence of a gene signature of anti-CD40 activated cDC1s previously defined in MC38 mouse tumors[52]. This analysis revealed a significant increase of the activated cDC1s gene signature compared to control only in tumors treated with RT+CTLA4i+anti-CD40 (Fig. 7a and b). Anti-CD40 also increased significantly AH1-specific TCR repertoire sharing between the tumor and draining lymph node (dLN) of each mouse as compared to controls (Fig. 7c), supporting the interpretation that the main effect of anti-CD40 is to increase cDC1 activation and tumor-specific T cell priming.

To further support this hypothesis, flow cytometry characterization of myeloid cells was performed after the first anti-CD40 administration. At this early time there was a reduction in total DCs in tumors treated with RT while anti-CD40 increased the proportion of cDC1 defined by XCR1 expression (Fig. 8a, b). The cDC1 population present in anti-CD40-treated tumors showed decreased expression of CD40, but increased expression of CD80 and CD86 (Fig. 8c–f). Interestingly, while anti-CD40 increased CD80 single-positive cDC1, RT increased CD40 and CD80 double-positive cDC1 (Supplementary Fig. 19a). In contrast, CD40 and CD86 double-positive cDC1 were relatively more represented after treatment with anti-CD40 (Supplementary Fig. 19b). RT significantly reduced intratumoral macrophages compared to control, largely at the expense of cells expressing CD206, a marker of M2 polarization (Fig. 8g, h). Expression of MHC-II and, to a lesser extent CD86, was reduced in the CD206$^-$ macrophage subset present in RT-treated tumors, while CD80 expression was unchanged (Supplementary Fig. 19c–e), suggesting that neither RT nor anti-CD40 induce macrophage activation, at least at this early time point.

In the dLN total DC and XCR1$^+$ cDC1 were increased in mice receiving anti-CD40, compared to control, accompanied by an increase in CD40 expression by cDC1 cells (Fig. 8i-l). All cDC1 in dLN were positive for CD80, CD86 and MHC-II and CD86 expression was increased by anti-CD40 (Fig. 8m, n). Anti-CD40 also increased CD11b$^+$ F4/80$^+$ macrophages in dLNs without altering the relative ratio of CD206$^+$ and CD206$^-$ macrophages (Fig. 8o, p).

Taken together, these results strongly suggest that the main effect of anti-CD40 in 4T1 tumors is to increase cDC1, promote their

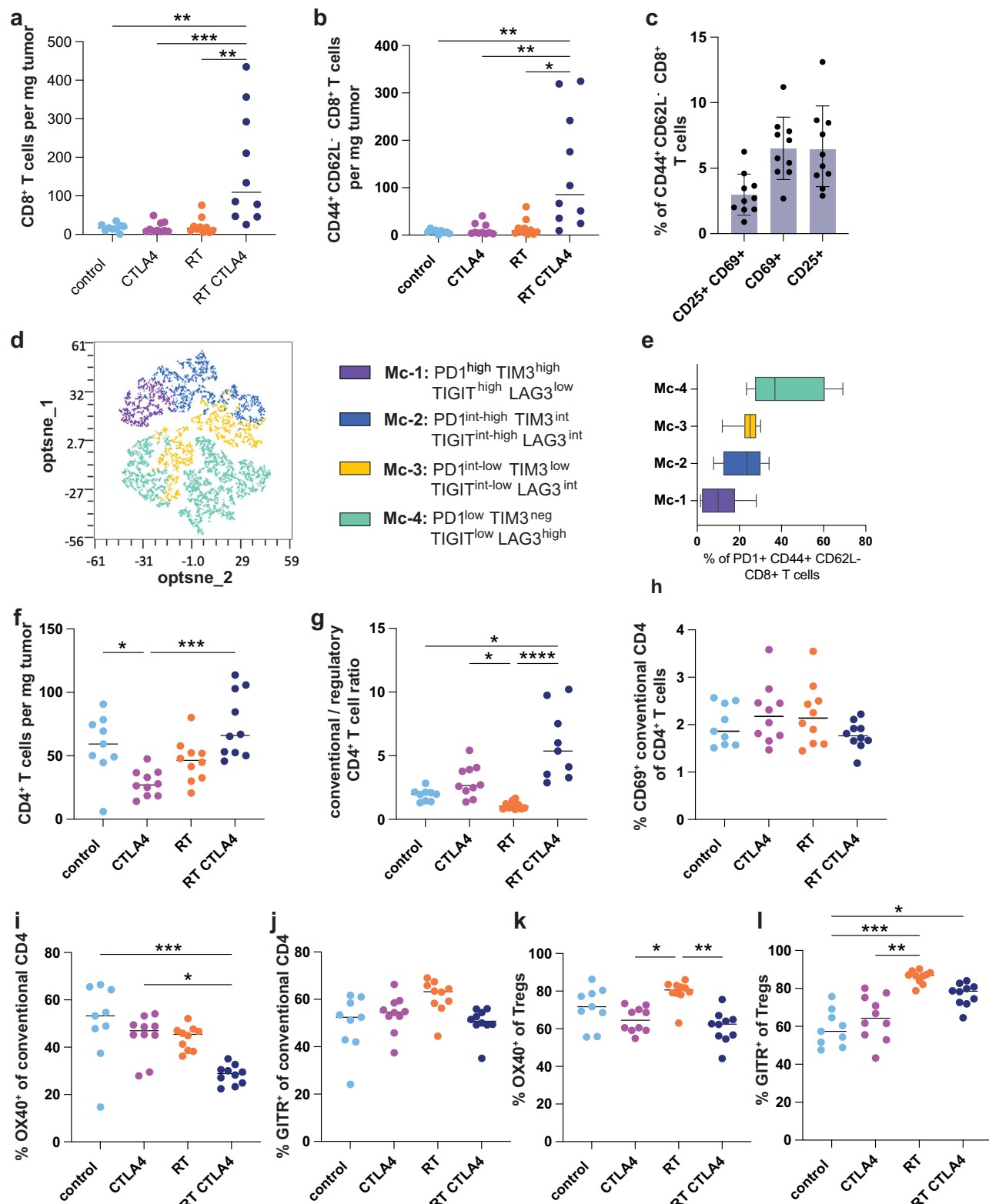

activation and migration to dLN where they cross-present tumor antigens to CD8 T cells, whereas RT reduces the presence of immunosuppressive macrophages in the tumor.

### Control of spontaneous lung metastases in mice treated with radiation therapy, CTLA4i and anti-CD40

Since the survival of 4T1 mice reflects not only control of the primary tumor but also of the lung metastases that develop spontaneously in this model, we investigated the effects of CD40 agonist on lung metastases and immune infiltrate in mice treated with RT and/or CTLA4i. Metastatic colonies were significantly reduced compared to control in animals treated with CD40 and CTLA4 antibodies used in combination with each other or with RT, with the most significant decrease observed in mice treated with RT+CTLA4i regardless of the addition of anti-CD40, while RT alone had no effect (Fig. 9a).

**Fig. 5 | RT + CTLA4i combination modulates the phenotype of intratumoral T cells.** BALB/c mice were injected with 4T1 tumors (n = 10 biologically independent mice/group except control n = 9), treated with RT (8 GyX3), CTLA4i or the combination as described in Supplementary Fig. 5. 2 days after the last dose of CTLA4i (day 22) tumors were collected and the immune infiltrate analyzed by flow cytometry. **a** Intra-tumoral density of CD8[+] T cells and (**b**) CD44[+] CD62L[-] antigen-experienced CD8[+] T cells. **c** Expression of activation markers by antigen-experienced CD8[+] T cells in RT + CTLA4i treated tumors. Data are mean ± SD. **d** High dimensional analysis of the CD8[+] T cell infiltrate of RT + CTLA4i treated tumors. CD8[+] CD44[+] CD62L[-] PD1[+] T cells were down sampled to 1000 cells per sample and concatenated. opt-sne was run using standard imputs (perplexity = 30, iterations = 1000) based on 4 markers (PD1, Lag3, TIGIT, Tim3). FlowSOM-based metaclusters

(Mc) are overlaid on the opt-sne 2D plot as a color dimension. **e** Frequency of the 4 metaclusters among antigen-experienced-PD1[+] CD8[+] T cells. Boxplots show the median and interquartile intervals, whiskers indicate minimum and maximum values of 8 tumors from the RT + CTLA4i group. **f** Intra-tumoral density of CD4[+] T cells; (**g**) conventional over regulatory (Foxp3[+] CD25[+]) CD4[+] T cells ratio. **h** Frequency of activated (CD69[+]) conventional CD4[+] T cells out of intra-tumoral CD4[+] T cells. **i–l** Percentage of OX40[+] and GITR[+] conventional (**h**, **i**) and regulatory (**j**, **k**) CD4[+] T cells. Lines indicate median and each dot represent one animal. Kruskal–Wallis test and post-hoc Dunn's test were performed on each individual panel *, **, ***, and ****, indicate p-values < 0.05, 0.01, 0.001, and 0.0001, respectively. Source data and exact p values are provided in the Source Data file.

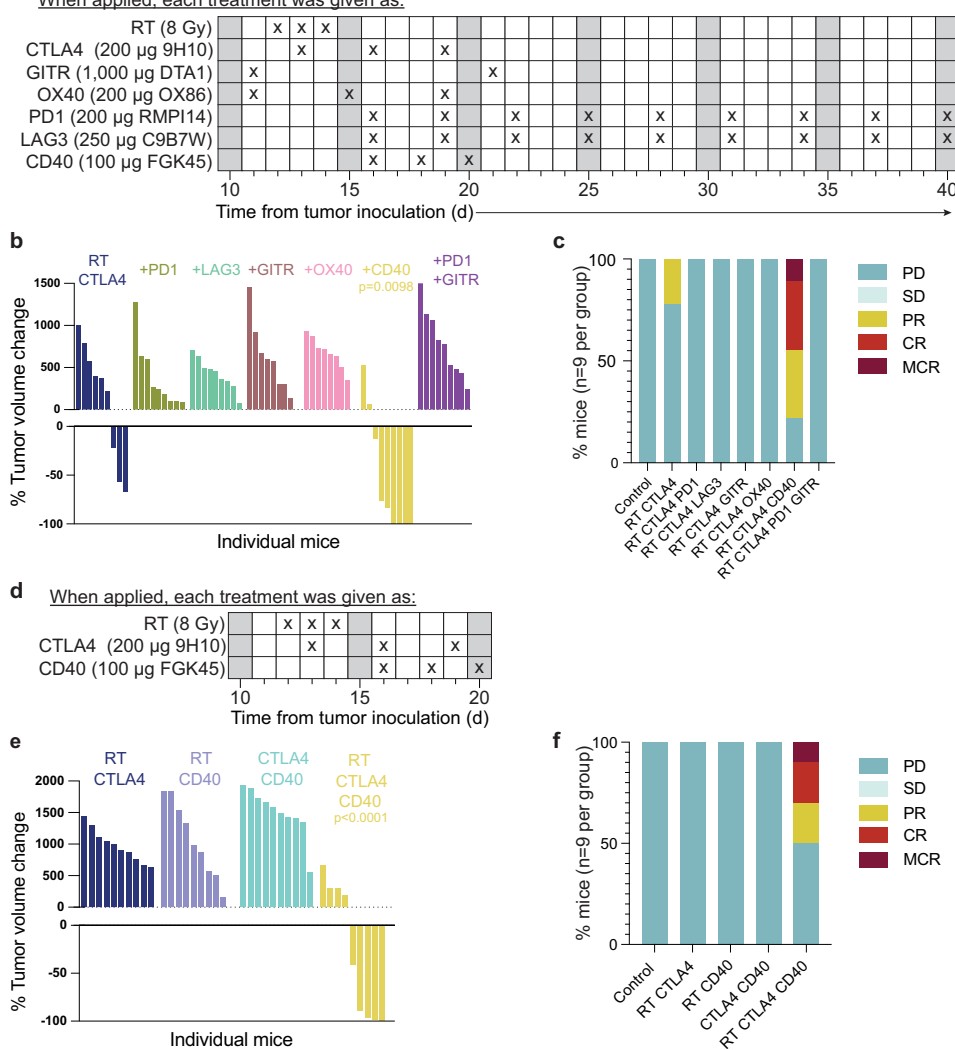

**Fig. 6 | Agonistic CD40 treatment improves 4T1 tumor response to RT + CTLA4i.** BALB/c female mice were injected s.c. with 4T1 cells at day 0 in the flank. **a** Treatment schedule for testing multiple combination therapies (**b**, **c**). Dosing and antibody clone name are indicated for each target tested in combination with RT+CTLA4i therapy, (n = 9 biologically independent mice/group except for control n = 5). **b** Waterfall plots showing tumor volume change between day 11 and day 33 following tumor inoculation. **c** Response to treatment of each mouse was measured using the 5-category method that classifies responses into complete response (CR), maintained CR (MCR), partial response (PR), stable disease (SD), and progressive

disease (PD). CR, complete tumor regression at at least one assessment with regrowth before the end of the experiment; MCR, remained tumor free until the end of the experiment. **d** Treatment schedule for **e** and **f**, (n = 10 biologically independent mice/group except for control n = 5). **e** Waterfall plots showing the best tumor response (day 31 following tumor inoculation). **f** Response to treatment combinations assessed using the 5-category method. Mann–Whitney test was applied to log-transformed tumor volume change values. Source data are provided in the Source Data file.

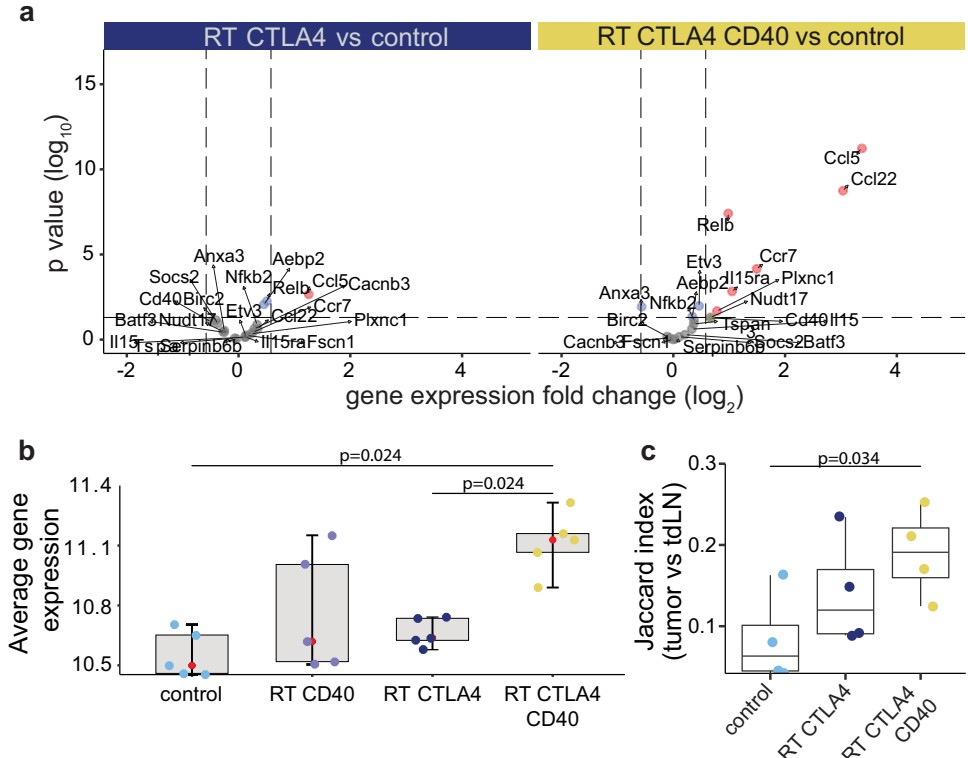

**Fig. 7 | Agonistic CD40 treatment increases cDC1 activation in RT + CTLA4i treated 4T1 tumors, and the frequency of AH1-clonotypes shared between tumor and dLN.** BALB/c mice implanted with 4T1 tumors were treated with the schedule in Fig. 6d. Tumor and tumor-draining lymph node (tdLN) were collected at day 22 for analysis of gene expression (tumor, *n* = 5 biologically independent mice/group, **a** and **b**) and TCR repertoire (tumor and tdLN, *n* = 4 biologically independent mice/group, **c**). **a** Volcano plots comparing the tumor expression of a gene signature of anti-CD40 activated cDC1[52] (Relb, Etv3, Batf3, Aebp2, Nfkb2, Ccl22, Ccl5, Il15, Ccr7, Il15ra, Plxnc1, Pmp, Cd40, Birc2, Fscn1, Anxa3, Cacnb3, Nudt17, Socs2, Tspan3, Serpinb6b) in untreated versus RT+CTLA4i (left) and RT +CTLA4i+anti-CD40-treated (right) 4T1 tumors. **b** The average expression

(generated via DESeq2) of the genes in the signature was determined as the arithmetic mean (red dot) of the log scale gene expression data in all treatment groups. A two-sided *t* test was used to evaluate statistical significance. **c** The Jaccard overlap index of the anti-tumor AH1-specific TCR repertoire between paired tumor and tdLN for each mouse was calculated to assess overlap between the two compartments. A Wilcoxon two-sided test was used to evaluate statistical significance. In (**b**, **c**) ggplot2::geom_jitter t was used to visualize individual values within each group. Boxplots display median (center line), 25th and 75th percentile (hinges) and minimum and maximum values (whiskers). Source data are provided as a Source Data file.

Anti-CD40 tended to reduce the relative proportion of total and CD8[+] T cells among lung CD45[+] cells and this reduction was significant when comparing anti-CD40 + CTLA4i to RT+CTLA4i (Fig. 9b, c). However, when added to RT and RT+CTLA4i, anti-CD40 significantly increased the percentage of CD69[+] CD8[+] T cells (Fig. 9d), a marker associated with activation and tissue residency of memory T cells[53]. The percentage of PD1[+] CD8[+] T cells was also increased compared to control by anti-CD40 treatment (Fig. 9e). In contrast, the percentage of AH1-specific CD8[+] T cells was increased in CTLA4i-treated mice compared to control, irrespective of anti-CD40 (Fig. 9f). Similarly to CD8[+] T cells, CD4[+] T cells were reduced among CD45[+] cells in mice treated with anti-CD40, but the percentage expressing CD69 and PD1 was significantly increased among conventional CD4[+] T cells compared to the treatment groups without anti-CD40 (Fig. 9g–i). Anti-CD40 also increased the percentage of CXCR3[+] conventional CD4[+] T cells in RT and RT+CTLA4i groups, although the increase was significant only when compared to RT+CTLA4i (Fig. 9j). Anti-CD40 increased the percentage of total DCs in all groups where it was included compared to control, but reduced the fraction positive for CD40 when used with CTLA4i or RT but not with RT+CTLA4i (Fig. 9k–m). Likewise, the XCR1[+] cDC1 subset was decreased when anti-CD40 was added to RT as compared to RT alone, but this effect was not seen when anti-CD40 was added to RT+CTLA4i (Fig. 9n). Macrophages were increased compared to control in the groups treated with RT+anti-CD40 and/or CTLA4i and appeared to be more activated in the triple combination

group compared to control based on expression of CD40 and MHC-II while CD80 showed minimal changes (Supplementary Fig. 20).

Overall, these results show that all combination therapies were able to reduce lung metastases. The triple combination of RT+CTLA4i +anti-CD40 was not superior to RT+CTLA4i, and metastases were reduced also when anti-CD40 and CTLA4i were used in combination but without RT. Given the increase in tumor antigen-specific CD8[+] T cells driven by CTLA4i, together with the increased expression of PD1 by lung CD4[+] and CD8[+] T cells driven by anti-CD40 we reasoned that anti-PD1 could be necessary to sustain responses elicited by RT +CTLA4i+anti-CD40. To our surprise anti-PD1 administered after completion of RT+CTLA4i+anti-CD40 did not have any beneficial effects (Supplementary Fig. 21).

## CD40 agonism enhances local and abscopal responses to radiation therapy and CTLA4i in mice bearing radiation and immunotherapy-resistant AT3 tumors

To determine the effect of anti-CD40 in another tumor model that is resistant to CTLA4i and radiation used as monotherapy and shows responses in a minority of mice treated with the combination of RT +CTLA4i (Fig. 10a–c, and Supplementary Fig. 22a), we used the AT3 mouse model of triple-negative breast cancer. AT3 has a tumor immune microenvironment dominated by myeloid cells, similarly to 4T1, but is derived from C57BL/6 rather than BALB/c mice and is only weakly metastatic compared to 4T1[54]. Mice treated with anti-CD40

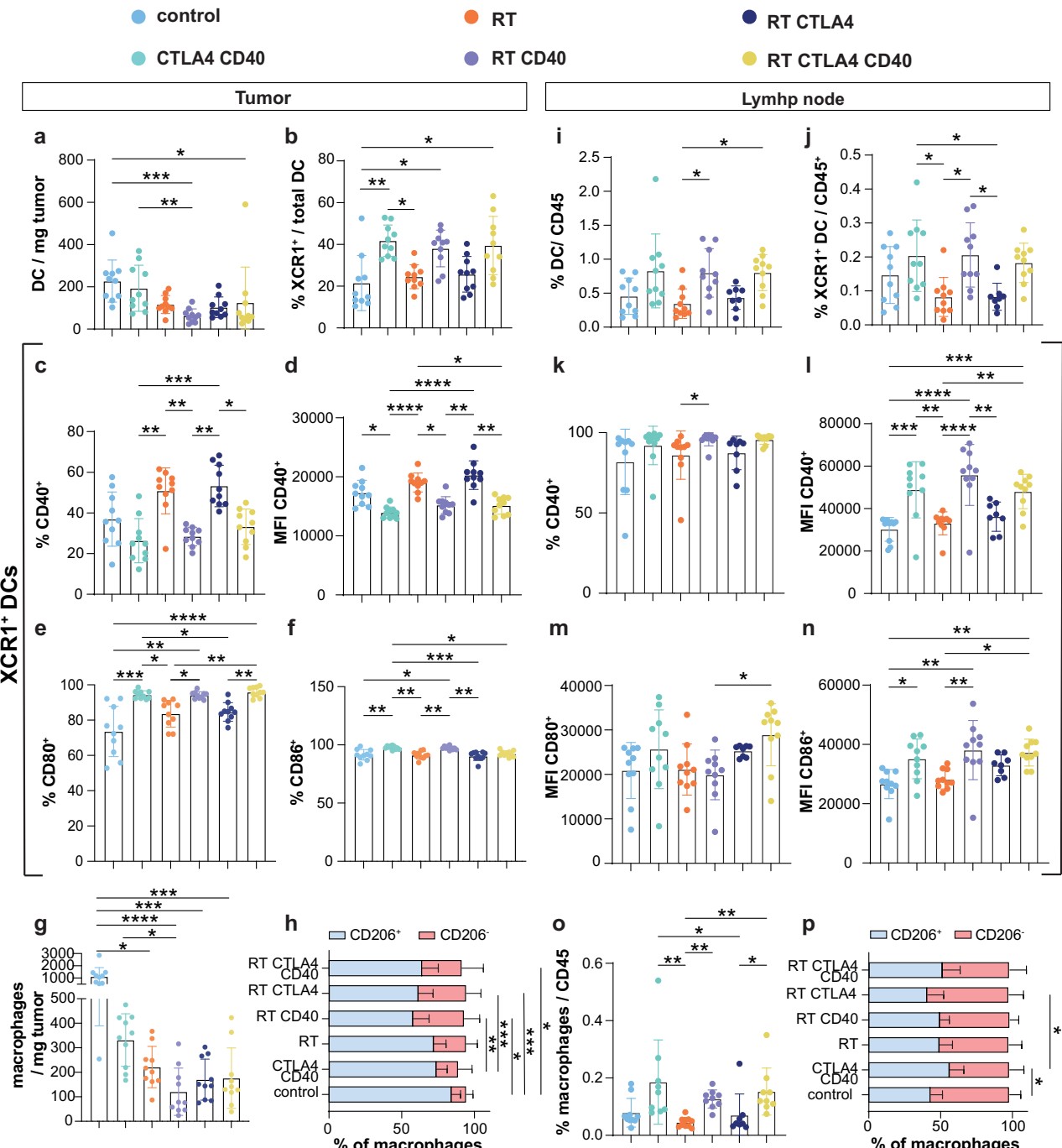

**Fig. 8 | Effects of anti-CD40 on the myeloid infiltrate of 4T1 tumor and tumor-draining lymph node.** BALB/c mice implanted with 4T1 tumors were treated with the schedule in Fig. 6d. Tumor (left panels) and tumor-draining lymph node (tdLN) (right panels) were collected at day 17 for analysis of myeloid cells by flow cytometry. Dendritic cells (DCs) were defined as CD11b⁺ F4/80⁻ CD11c⁺ cells among CD45⁺ cells (**a**, **i**), and DC1 were further defined by expression of XCR1 (**b–f** and **j–n**). Percentage of DC1 positive for CD40 (**c**, **k**) and mean fluorescence intensity (MFI) (**d**, **l**). Percentage of DC1 positive for CD80 (**e**) and CD86 (**f**) in tumor. Expression of CD80 (**m**) and CD86 (**n**) by DC1 in tdLN. Macrophages were defined as CD11b⁺ F4/80⁺ among CD45⁺ cells (**g**, **h** and **o**, **p**). M2-like macrophages were furher defined by expression of CD206. Treatment-related changes in their percentage in the tumor (**h**) and tdLN (**p**). Data are shown as mean ± SD, each dot represents one animal, n = 10 biologically independent mice/group. Populations were compared by Kruskal–Wallis and Dunn's post-test for statistical significance. *, **, *** and **** indicate p-values < 0.05, 0.01, 0.001 and 0.0001 respectively. Source data and exact p values are provided in the Source Data file.

alone or combined with CTLA4i did not show any response. When combined with radiation anti-CD40 slowed tumor progression without achieving a partial response. In contrast, the combination of RT+CTLA4i +anti-CD40 led to partial or complete tumor responses in 60% of the mice. Complete responses were durable and associated with increased survival and protective memory in mice treated with RT+CTLA4i or RT+CTLA4i+anti-CD40, but the addition of

anti-CD40 doubled the response rate (Fig. 10b–d and Supplementary Fig. 22a, b).

Next, to determine if RT+CTLA4i+anti-CD40 could induce responses against a non-irradiated tumor (abscopal response), mice were implanted with AT3 cells in both flanks, with radiation delivered only to one of the two tumors. Complete regression of the irradiated tumor was observed in 42% of the mice treated with RT

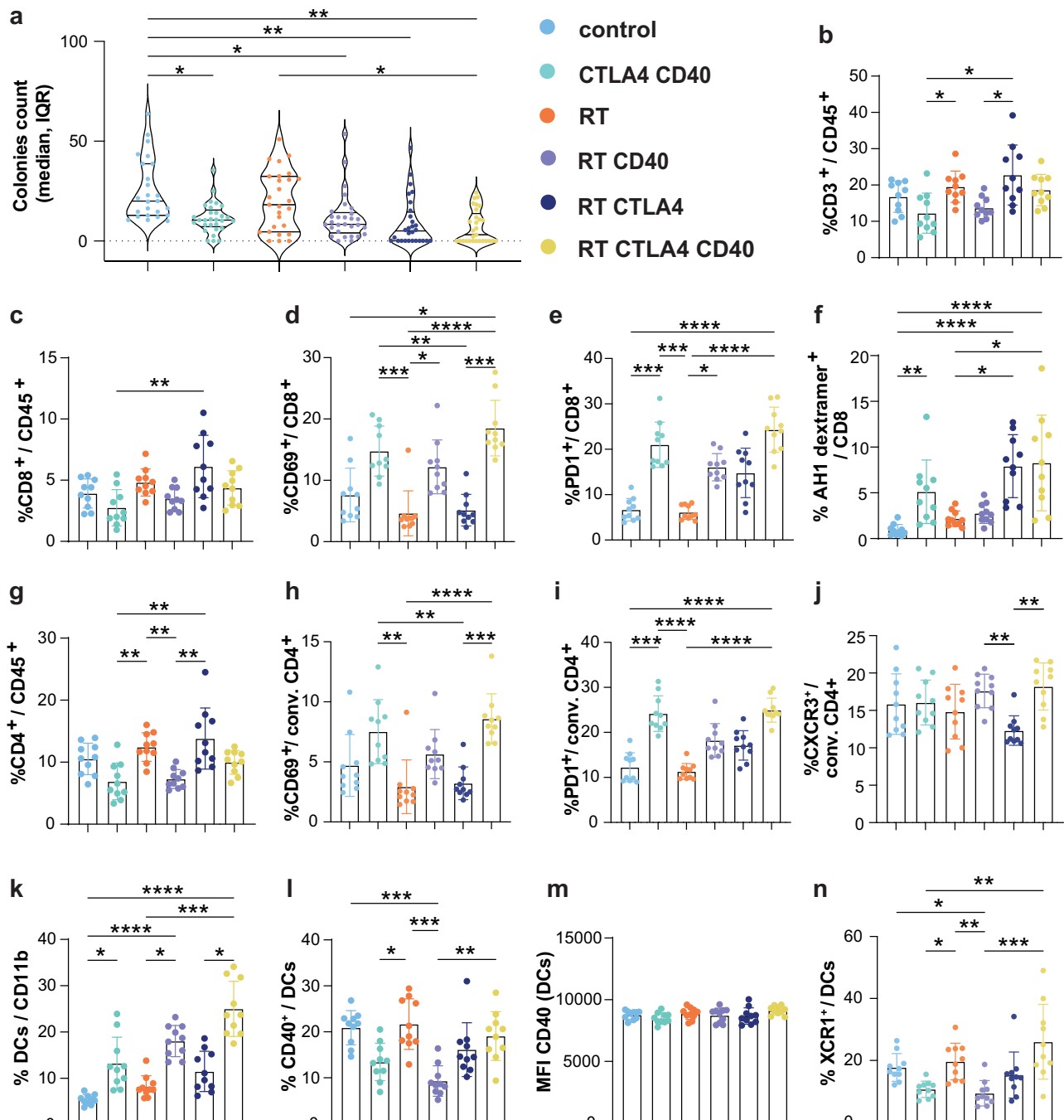

**Fig. 9 | Effects of RT, CTLA4i, and anti-CD40 on spontaneous 4T1 lung metastasis and lung immune cells.** BALB/c mice implanted with 4T1 tumors were treated with the schedule in Fig. 6. Lungs were collected at day 22 to evaluate the immune infiltrate and tumor cell content. **a** Number of clonogenic 4T1 lung metastasis at day 22. Each dot represents a single replicate (4 to 6 replicate cultures per mouse, 10 biologically independent mice per group). Square root transformed data were used to ensure the underlying model assumptions were satisfied. P-values were adjusted for multiple comparisons by controlling the false discovery rate. Horizontal lines indicate the median, Q1 and Q3. **b** Percentage of CD3⁺ T cells among CD45⁺ cells. **c** Percentage of CD8⁺ T cells among CD45⁺ cells and their expression of CD69 (**d**)

and PD1 (**e**). **f** Percentage of AH1-specific CD8⁺ T cells defined by dextramer staining. (**g**) Percentage of CD4⁺ T cells among CD45⁺ cells. Expression of CD69 (**h**) and PD1 (**i**) by conventional (conv) CD4⁺ T cells, defined as FOXP3⁻. **j** Percentage of CXCR3⁺ among conventional CD4⁺ T cells. **k** Percentage of DCs, defined as Ly6G⁻ CD11c⁺ F4/80⁻ among CD11b⁺ cells. Percentage of expression (**l**) and expression levels (**m**) of CD40 by DCs. **n** Percentage of XCR1⁺ DCs. **b**–**n** Data are shown as mean ± SD, each dot represents one animal, $n = 10$ biologically independent mice per group. Populations were compared by Kruskal–Wallis and Dunn's post-test for statistical significance. *, **, *** and **** indicate $p$-values < 0.05, 0.01, 0.001 and 0.0001 respectively. Source data and exact $p$ values are provided in the Source Data file.

+CTLA4i+anti-CD40 (Fig. 10e); highly significant delayed progression of the abscopal tumor was also observed in this group, with two partial responses and one complete response reflecting in increased survival (Fig. 10f–i). Complete response achieved in both, irradiated and abscopal tumor was durable lasting > 100 days. Thus, radiation

cooperated with CTLA4i and CD40 agonism to improve responses of both, irradiated and non-irradiated tumors.

Taken together, these results highlight the complexity of choosing effective combination therapies for immunotherapy-resistant tumors and suggest that targeting complementary immune cell

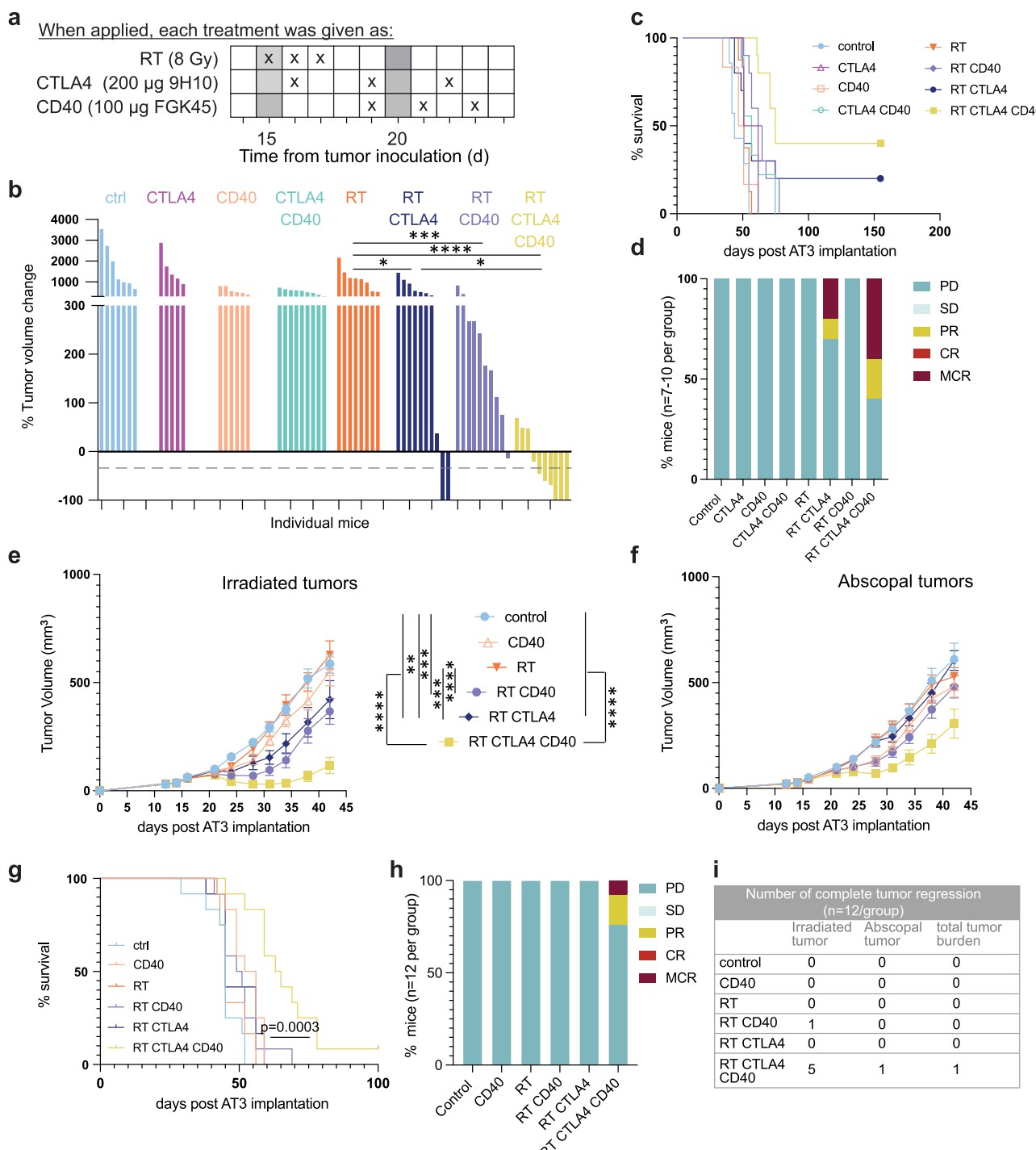

**Fig. 10 | Agonistic CD40 treatment improves AT3 tumor response to RT + CTLA4i.** C57BL/6 female mice were injected s.c. with AT3 cells at day 0 in the flank. **a** Treatment schedule. **b** Waterfall plots of tumor volume change between day 15 and day 35 post-tumor inoculation. Treatment groups were compared by Mann–Whitney tests on Log-transformed values for statistical significance. All statistical tests were two-sided. **c** Survival (n = 10 biologically independent mice/ group except for control n = 5). **d** Response to treatment of each mouse measured using the 5-category method, as in Fig. 6. **e** Mice were injected with AT3 tumors in both flanks, radiation was given only to one tumor and CTLA4i and/or anti-CD40 were given according to the schema in (**a**) (n = 12 mice per group). Tumor growth curves for irradiated (**e**) and abscopal (**f**) tumors, mean ±SEM. Statistical

significance was assessed by repeated-measures 2-way ANOVA. **g** Survival (p < 0.001; log rank test). **h** Relative tumor volume was calculated using the total (primary + abscopal) tumor volume in comparison to the total tumor volume at day of treatment start. Response to treatment was assessed using the 5-cat method. Maintained complete responders rejected both irradiated and abscopal tumors and remained tumor-free until the end of the experiment. **i** Complete tumor regression observed on primary/secondary or both tumors for each treatment combination group. *,**, ***, and ****, indicate p-values < 0,05, ≤0.005, ≤0.001, and <0.0001, respectively Source data and exact p values are provided in the Source Data file.

compartments is necessary but not sufficient in the absence of radiation to achieve responses of established tumors, whereas it may be sufficient for the control of micro-metastases.

## Discussion

The combination of radiation with CTLA4i has been tested in multiple clinical studies and different tumor types, with encouraging results but overall little evidence of a consistent benefit[6–8,55,56]. Progress in improving the effectiveness of this combination is hindered by a lack of understanding of the mechanisms whereby these therapies interact to generate effective T-cell responses. Here we performed an in-depth analysis of the T cell response shaped by CTLA4i in irradiated and non-irradiated tumors using bulk and single-cell RNA sequencing in the mouse carcinoma 4T1, which is completely resistant to CTLA4i monotherapy but responsive to the combination of radiation and CTLA4i[3]. We found that CTLA4i by itself was unable to increase the baseline T cell infiltration in 4T1 tumors, but it shifted the balance between different T cell subsets, increasing the frequency of $CD4_{TH1}$-like cells largely at the expense of $CD8_{EX}$ cells. These data mirror reports in patients showing that anti-CTLA4 enhances $CD4_{TH1}$ and, to a lesser degree, $CD8^+$ T cell responses[57]. Although the anti-CTLA4 antibody used, clone 9H10, has been shown to deplete intratumoral $CD4_{TREG}$[58], the latter were only minimally affected in 4T1 tumors treated with CTLA4i alone, but where significantly decreased in mice treated with RT+CTLA4i. It is possible that RT favors $CD4_{TREG}$ depletion by increasing the availability of macrophages expressing activating FcγRIV[58]. Alternatively, RT-induced elimination of carcinoma cells could increase glucose availability in the tumor microenvironment, making $CD4_{TREG}$ susceptible to CTLA4i-induced phenotypic and functional destabilization[12]. Our studies were not designed to directly test the role of $CD4_{TREG}$ in the tumor response to RT+CTLA4i, and despite the lack of effectiveness of antibodies targeting receptors highly expressed by intratumoral $CD4_{TREG}$ such as GITR and OX40, we cannot rule out a contribution of these cells to the therapeutic resistance to RT+CTLA4i.

Radiation decreased the relative proportion of all $CD4^+$ T cell subsets while increasing the overall T cell infiltration in the tumor, and this may explain, in part, the strong effect of radiation on TCR clonality, as $CD8^+$ T cells were generally more clonal than $CD4^+$ T cells. Many $CD8^+$ T cells were specific for the known epitope AH1, derived from the envelope protein of an endogenous retrovirus. Although the AH1-specific TCR motif used here was previously defined in a comprehensive analysis of the AH1-specific T cell response[9], we recognize that inferring antigen specificity based on the beta chain alone, as we did in the bulk TCR analyses, may have some limitations. However, in the single cell analysis experiment AH1-specific T cells were identified by dextramer staining. Notably, in untreated tumors AH1-specific $CD8^+$ T cells were virtually absent among $CD8_{EA}$ and $CD8_{EM}$/C1, while they became prominent in RT+CTLA4i treated tumors, representing the majority of $CD8_{EA}$. However, the dominant $CD8^+$ T cell clones were not AH1-specific. We have recently identified a mutational neoantigen recognized by $CD8^+$ T cells that is presented in markedly higher amounts in irradiated as compared to untreated 4T1 cells[59], suggesting that such neoantigen(s) could be the target(s) of $CD8^+$ T cells that infiltrate irradiated tumors. Further investigations are needed to identify the antigenic targets of the $CD8^+$ T cells in 4T1 tumor and determine if they are different in irradiated and untreated tumors.

The combination of RT and CTLA4i reprogrammed the T cell landscape by leading to the expansion of $CD8_{PEX}$, $CD8_{EA}$, $CD8_{EM}$, in addition to expanding activated $CD4_{TH1}$ cells and reducing $CD4_{TREG}$ cells. Overall, these changes are consistent with the therapeutic synergy of this combination, which also markedly increased intratumoral T cells, effectively converting a T cell-poor 4T1 tumor into a T-cell-inflamed tumor. In patients, such microenvironment reflects a spontaneous anti-tumor immune response that is associated with improved survival in multiple cancer types and is a predictor of response to immune checkpoint inhibitors[60].

We found that a combined gene signature that incorporated the three clusters that were uniquely enriched in tumors after RT+CTLA4i therapy ($CD8_{EM}$/C1, $CD8_{EA}$, and $CD8_{PEX}$) was associated with improved survival in patients with triple-negative breast cancer and melanoma, independently from total T cell infiltration, suggesting that these clusters are functionally significant. Thus, the combination of RT with CTLA4i can convert a tumor with little T cell infiltration into a highly T-cell-inflamed tumor, dominated by T-cell differentiation states associated with anti-tumor activity.

Despite this, in the 4T1 tumor the increased tumor infiltration by effector T cells did not set the stage for response to anti-PD1 or other antibodies targeting the T cell immunomodulatory receptors GITR, OX40, and LAG3. These results are in contrast with a prior report in the mouse B16 melanoma that showed improved responses when PD1 blockade was added to radiation and CTLA4i[8]. It is likely that this difference reflects the distinct immune contexture of B16 and 4T1 tumors[61]. A recently reported phase I study of melanoma patients treated with the combination of nivolumab, ipilimumab, and radiotherapy did not reproduce the improved systemic responses observed in B16, but control of the irradiated lesions was better than expected with radiation alone[62], suggesting some efficacy, although a randomized trial is required to dissect the contribution of each immune checkpoint inhibitor.

In stark contrast to the antibodies targeting T cell immunomodulatory receptors, agonist anti-CD40 markedly improved tumor regression in mice treated with RT+CTLA4i. The main mechanism of action of agonist CD40 antibodies is the activation of DCs and other antigen-presenting cells[50], thus targeting a cellular compartment that is essential not only for priming and activation of T cells, but also to support the functionality of T cells in the tumor microenvironment[63]. Consistently, our data show that anti-CD40 main effect was to activate cross-presenting DCs at the irradiated tumor site and draining lymph node and promote the expansion of tumor-specific $CD8^+$ T cells migrating between the dLN and the tumor in mice treated with RT+CTLA4i. The need to provide additional CD40 agonism may be explained by the low expression of CD40LG on $CD4_{TH1}$ cells in tumors treated with RT+CTLA4i, which suggests that CTLA4i drives the expansion of a population of effector $CD4^+$ T cells that are somewhat defective in their helper function.

CD40 antibodies have been shown in preclinical studies to improve responses to combinations of chemotherapy or radiotherapy with anti-PD1 and anti-CTLA4 in pancreatic cancer, one of the tumors most resistant to immunotherapy[64,65]. Clinical studies testing different CD40 agonists in a variety of tumor types have yielded encouraging but inconclusive results[50]. Following encouraging results of a phase 1b trial in metastatic pancreatic cancer of chemotherapy with CD40 agonist[66], a phase II trial was recently performed, which compared chemotherapy with anti-PD1, anti-CD40, and their combination. Interestingly, there was no obvious benefit of the addition of anti-PD1 to chemotherapy with CD40 agonist[67]. An exploratory analysis of biomarkers predictive of response highlighted differences between patients who responded to chemotherapy+anti-PD1 versus chemotherapy+anti-CD40, with CD4 T cells and antigen-presenting cells associated with longer survival to chemotherapy+anti-CD40[67]. Based on the importance of cross-priming DC in the activation of anti-tumor T cells, a recently opened trial will test the combination of Flt3 ligand, a growth factor for DC, anti-CD40 and chemotherapy in patients with metastatic TNBC[68]. Given the ability of RT to recruit DCs via IFN type I[69], and the results of the preclinical studies reported here in two TNBC models, it is intriguing to consider whether radiation with CTLA4i and anti-CD40 should be considered as alternative modalities for testing in this and other diseases with poor response to immunotherapy.

Whereas the addition of anti-CD40 dramatically improved the rejection of the irradiated tumor it did not further improve the inhibition of lung metastases achieved by RT+CTLA4i in 4T1 bearing mice. In contrast, the addition of anti-CD40 was required to achieve control of a non-irradiated subcutaneous tumor in AT3-bearing mice treated with RT+CTLA4i. These results suggest that therapeutic CD40 agonism may be required for the effectiveness of RT+CTLA4i in established tumors that are dominated by an immune suppressive myeloid infiltrate such as 4T1 and AT3[54]. In contrast, 4T1 lung micro-metastases were reduced by all combination therapies tested, including CTLA4i+anti-CD40 used without RT, but no treatment was able to completely eliminate metastatic cells in the majority of the mice. These data raise the question whether mechansims of tumor resistance to immune-mediated rejection are different in different organ sites, implying the need for different therapeutic strategies.

In summary, we performed an in-depth analysis of the effects of radiation and CTLA4i on the functional differentiation of intratumoral T cells in an aggressive and immunotherapy-refractory mouse carcinoma. Our data provide insights into the mechanisms of synergy of these therapies by showing that only when used together they lead to the emergence of CD8 functional subsets that are associated with increased survival in patients, accompanied by a decrease in CD4 regulatory T cells. Results of our studies also suggest that the expression of a costimulatory or coinhibitory receptor by intratumoral T cells does not always predict the efficacy of cognate therapeutic antibodies, and support combination treatments that target complementary immune cell subsets.

## Methods

### Ethical statement

All mouse experiments were approved by the Institutional Animal Care and Use Committee at Weill Cornell Medicine.

**Cell lines.** 4T1 cells were obtained in 2001 from Fred R. Miller of Karmanos Cancer Institute, who established this mammary carcinoma cell line[70], and a large stock of low passage frozen cells prepared. Cells from this stock were authenticated by IDEXX Bioresearch (Columbia, MO, USA) in 2019 by genetic evaluation of interspecies contamination and mouse STR profile. AT-3 cells were obtained from J Schlom and authenticated by IDEXX Bioresearch (Columbia, MO, USA) in 2016. Cells were further authenticated by morphology, growth and pattern of metastasis in vivo and routinely screened for Mycoplasma (LookOut Mycoplasma PCR Detection kit, Sigma-Aldrich, St. Louis, MI). Cells were maintained in DMEM (Invitrogen) supplemented with 2mmol/L Lglutamin, 100 U/mL penicillin, 100 μg/mL streptomycin, 2.5 × $10^{-5}$ mol/L 2-mercaptoethanol, and 10% FBS (Life Technologies). 4T1 and AT-3 cells were routinely cultured for less than a week before injection into the mice.

**Interferon beta ELISA measurements.** 4T1 cells were seeded into 28.2 $cm^2$ dishes and irradiated with single doses of 4, 6, 8, 12, 16, 20, and 24 Gy or repeated doses of 8, 6 and 4 Gy on 3, 4 and 6 consecutive days, respectively. Cell-free supernatant was collected 24 hours after the last dose and IFNB1 was measured using the VeriKine-HS Mouse Interferon Beta ELISA Kit (PBL Assay Science), according to the manufacturer's instruction. 50 μL of undiluted supernatant or recombinant IFNB1 standards were processed in technical triplicate wells. 450 nm optical density was measured using a FlexStation 3 plate reader (Molecular Devices). Concentrations were normalized by the number of viable cells.

**Tumor growth and treatment.** Six to eight-week-old female BALB/c and C57Bl/6 mice were obtained from Taconic (Germantown, NY) and Jackson Laboratory (Bar Harbor, ME). The animal holding room is maintained at 72 ± 2 °F (21.5 ± 1 °C), relative humidity between 30% and 70%, and a 12:12 hour light:dark photoperiod. Only female mice were implanted with 4T1 and AT3 cells since these are models of breast cancer that is very rare in males. Mice were subcutaneously inoculated with $5 \times 10^4$ 4T1 cells or $5 \times 10^5$ AT3 cells in one or both flanks. Tumor growth was monitored two to three times a week using a Vernier caliper and tumor volume was calculated using the formula: length x width$^2$ x p/6. Maximum tumor volume as per our Institutional Animal Care and Use Committee approved protocol is 1500mm$^3$, and mice were euthanized when tumor reached this volume. For pre- and post-treatment comparison, mice were inoculated in both flanks and one of the two tumors was removed surgically before the start of treatment. When the tumors reached 60-70mm$^3$, usually at day 11 (4T1) or 15 (AT3) post-inoculation, the mice were randomized to the different treatment groups. For conformal tumor irradiation all mice were anesthetized and either mock-treated or treated with 8 Gy dose of radiation on three consecutive days using the Small Animal Radiation Research Platform (SARRP Xstrahl Ltd, Surrey, UK). Anti-CTLA4 antibody (clone 9H10, BioXcell, West Lebanon, NH, USA) was given intraperitoneally in 100 μl PBS at a dose of 200 μg/injection, every 3 days starting on the second day of radiation. For single-cell sequencing, TCR sequencing, and flow cytometry analysis tumors and/or lungs were harvested at day 22.

In experiments comparing multiple treatment arms for the ability to induce tumor control mice were randomly assigned to the different treatment groups and treated with focal radiation therapy and anti-CTLA4 with or without administration of anti-GITR (clone DTA1, bioXcell, 1 mg/injection every 10 days, twice), anti-OX40 (clone OX-86, BioXcell, 200 μg/injection every 4 days, 3 times) starting 1 day prior to the first RT dose[45,48,71]. Anti-PD1 (clone RMP1-14, BioXcell, 200 μg/injection) and anti-LAG3 (clone C9B7W, BioXcell, 250 μg/injection), were started 2 days after the last dose of RT (d16) and were maintained every 3 days until 3 consecutive increasing tumor volumes were recorded for each animal in the group[44]. Agonist anti-CD40 (clone FGK45, BioXcell, 100 μg/injection) treatment was also initiated after RT at day16 and 3 doses were given every other day[64]. CD8 depletion was performed using the 2.43 anti-CD8 clone (BioXcell, 200 μg/injection); depletion was started 2 days before agonist anti-CD40 and maintained once a week. All antibodies were given i.p. in 100 μl PBS.

Response to combination treatments was assessed using the 5-cat method[72]. Relative tumor volume was calculated using randomization day (one day before RT#1$-V_{ref}$) as a reference using the formula: ($V_t \times (100/V_{ref}))/100$ with $V_t$ = tumor volume at each time point. Response groups were defined as follows: RTV > 0.5 during the study period and >1.25 at the end of the study: Progressive Disease (PD); RTV > 0.5 during the study period and ≤1.25 at the end of the study: Stable Disease (SD); 0 < RTV ≤ 0.5 on at least one measurement: Partial Response (PR); RTV = 0 on at least one measurement: Complete Response (CR); RTV = 0 at the end of the study: Maintained CR (MCR). Mice showing complete regression that remained tumor-free 100 day post-tumor injection (end of study point) were rechallenged with 4T1 or AT3 cells respectively on the contra-lateral side.

### Surgery

Mice were anesthetized in an isoflurane chamber, positioned over a heating pad and kept under anesthesia with isoflurane nose cone for the duration of the procedure. The areas were shaved and sterilized using 10% povidone-iodine (Betadine) and 70% ethanol and an eye ointment was applied to prevent dryness. Surgical resection of tumor samples was performed under both systemic (isoflurane) and local anesthesia (Bupivacaine), as well as local analgesia (Meloxicam injected s.c.) to the area of incision. The incision was closed using sterilized wound clips (Autoclips). Following surgery, moist chow and diet gel was provided ad libitum. The mice were given local analgesia (Meloxicam) as needed following surgery, and at least daily for 48 hrs. For all animal experiments, tumor growth was measured at least twice

a week and mice were sacrificed based on a predefined set of criteria per protocol.

**Isolations of genomic DNA, total RNA, and mRNA used for bulk sequencing.** Prior to DNA and RNA isolations, snap-frozen whole tumors were minced using a TissueRuptor (Qiagen). Then, total RNA and genomic DNA were isolated from the tumor lysate using AllPrep DNA/RNA Mini Kit (Qiagen), and mRNA was isolated from total RNA using the Dynabeads mRNA DIRECT Purification Kit (Invitrogen). All isolations were performed according to the manufacturer's instructions.

**Bulk tumor RNA sequencing.** Preparation of RNA sample library and RNA-seq was performed by the Genomics Core Laboratory at Weill Cornell Medicine. Messenger RNA was prepared using TruSeq Stranded mRNA Sample Library Preparation kit (Illumina, San Diego, CA), according to the manufacturer's instructions. The normalized cDNA libraries were pooled and sequenced on Illumina HiSeq4000 sequencer with pair-end 75 cycles. Illumina bcl2fastq2 v2.20 conversion software was used to demultiplex samples into individual sample and converted per-cycle BCL base call files into FASTQ files for downstream data analysis. The sequencing reads were cleaned by trimming adapter sequences and low quality bases *cutadapt* v1.9.1[73], and were aligned to the mouse reference genome (GRCm38) using *STAR* v2.5.2b[74]. Raw read counts per gene were extracted using *HTSeq-count* v0.11.2[75]. Differential expression analysis was performed using *DESeq2* v1.22.2[76], with significance cutoffs: *p*-value < 0.01 and |fold change| > 2.

**Bulk tumor TCR sequencing.** For TCRα and TCRβ CDR3 region sequencing, libraries were amplified from an average of 120 ng mRNA using the commercially available amplicon rescued multiplex polymerase chain reaction (arm-PCR) technology (iRepertoire), according to manufacturer's instructions[77]. Briefly, the arm-PCR technology is a multiplex amplification strategy that uses two PCR reactions to amplify the TCR repertoire. In the first reaction, nested TCRα and TCRβ V- and C-gene specific primers were used for reverse-transcriptase PCR. The primers also include sequencing adaptors for the Illumina platforms and barcodes used for downstream demultiplexing of sample-specific TCR libraries. In the second reaction, communal sequencing primers were used to exponentially amplify the product from the first PCR reaction. Following this, the concentration of the final TCR library product was measured using the Agilent High Sensitivity DNA kit and the Qubit 3.0 system, and library amplification was considered successful if the DNA concentration was > 10 ng/μl. After PicoGreen quantification and quality control by Agilent BioAnalyzer, TCR libraries were run on either a MiSeq (8 libraries; MiSeq Reagent Kit v3, 600 Cycles; Illumina) or HiSeq 2500 in Rapid Mode (92 libraries; HiSeq Rapid SBS Kit v2; Illumina) in a 250 bp/250 bp paired end run. The loading concentrations and PhiX spike-ins (to increase diversity and for quality control purposes) for MiSeq and HiSeq was 6pM and 5pM and 10% and 20%, respectively. The single MiSeq run yielded ~26 M reads, whereas the HiSeq 2500 runs yielded on average 360 M reads. The fastq files were submitted to iRepertoire for demultiplexing, and eventually, downstream analysis was performed using library specific tab-separated value (tsv) files. Non-productive TCRs were removed, and only productive nucleotide rearrangements and those with a frequency > $2 \times 10^{-5}$ was used for downstream analysis. For three mice treated with radiation monotherapy whose TCR repertoires was assessed using the iRepertoire platform, amplification and sequencing of TCRB CDR3 regions were also performed using the ImmunoSEQ platform at Adaptive Biotechnologies (Seattle, WA), as previously described[9,78]. This was performed to assess any difference between ImmunoSEQ and iRepertoire platforms (Supplementary Fig. 23). Lastly, the ImmunoSEQ platform was also used to assess the TCR

repertoire in paired tumor draining lymph nodes and tumors from untreated and mice treated with RT+CTLA4i or RT+CTLA4i+ anti-CD40 therapy. Related to the TCR repertoire analysis, clonality was calculated as:

$$\text{Clonality} = 1 - \frac{\text{Shannon Entropy}}{\log_2(n)}, \text{where} \quad (1)$$

$$\text{Shannon Entropy} = -\sum_i f_i \times \log_2(f_i), \quad (2)$$

and where $n$ = number of unique clones and $f_i$ = frequency of clone $i$. Jensen–Shannon Divergence, *JSD*, was calculated as

$$JSD(TCR_1||TCR_2) = \frac{1}{2}KLD\left(TCR_1||\frac{TCR_1+TCR_2}{2}\right) + \frac{1}{2}KLD\left(TCR_2||\frac{TCR_1+TCR_2}{2}\right), \text{where} \quad (3)$$

$$KLD(P||Q) = \sum_i P_i \ln \frac{P_i}{Q_i}, \quad (4)$$

and where *TCR1* and *TCR2* are two different TCR repertoires to be compared, *KLD* are Kullback–Leibler divergence, $P_i$ and $Q_i$ are frequencies of clone $i$ in repertoires $P$ and $Q$, respectively. Calculations of TCR repertoire statistics and comparisons were performed using a combination of functions in the *divo*, *tcR*, and *immunarch* packages in R. Data handling and post-processing and production of figures were performed using the *tidyverse* package and accompanying packages.

**Single-cell sequencing and analysis.** For in-depth profiling 4T1 infiltrating T cells, we utilized the 10x Genomics 5′ gene expression, V(D)J, and dCODE dextramer single-cell sequencing platform. Tumors were digested on a gentleMACS Dissociator using the mouse Tumor Dissociation Kit (Miltenyi Biotec), according to the manufacturer's instructions. To reduce batch effects, each digestion run always included tumors from all treatment groups. Directly after washing step following tumor digestion, cells were stained with fluorochrome-conjugated antibodies against CD3E (BV-650, clone 17A2, Biolegend), CD8a (BV-785, clone 53.6.7, Biolegend), CD4 (FITC, GK1.5, Biolegend), and Cd11b (AF-700, clone M1/70, Biolegend), and 10x Genomics compatible PE-labeled DNA-barcoded (CAAGCCACTGCTCC) dCODE H2-L$^d$ MHC class I dextramers linked to the AH1 peptide (SPSYVYHQF) (Immudex), according to manufacturer's instructions. To measure viability, DAPI was used. Directly after staining, CD3$^+$CD4$^+$ or CD3$^+$CD8$^+$ T cells were sorted on a BD FACSAria-II (BD Biosciences). In all, 20,000 cells were sorted from each tumor (n$_{tumors}$=5/condition) and pooled according to condition. Each cellular suspension (74–89% viability at concentrations between 590-980 cells/μl) was loaded onto the 10x Genomics Chromium Controller to partition single cells in Gel Beads-in-Emulsion (GEM), targeting about 5000 single cells per sample. Within each GEM, incubation with reverse transcriptase and poly dT primers generated first strand cDNA from the polyA RNA which was 5′ barcoded from the Gel Bead primers which contain a 10x cell barcode (16nt), a unique molecular identifier (UMI, 10nt) and a 13nt template switch oligo (TSO) (53 °C for 45 min in a C1000 Touch Thermal cycler with 96-Deep Well Reaction Module Bio-Rad, Hercules). Simultaneously, in the same partition, for those cells which have captured the barcoded dextramer, the incubation with reverse transcriptase and Gel Bead primers resulted in a 10 × 5′ barcoded oligonucleotide. GEMs were broken and DNA cleaned up with DynaBeads MyOne Silane (Thermo Fisher Scientific, Waltham, MA). cDNA and the dextramer oligonucleotide DNA were amplified with 14 cycles of PCR (10x Genomics, PN-2000119; 98 °C for 45 s; 98 °C for 20 s, 67 °C for 30 s, 72 °C for 1 h). Amplification products were cleaned up with SPRIselect beads (Beckman Coulter, Indianapolis, IN), and the cDNA fraction

(mean size of 750 bp) was separated by size selection from the dextramer oligonucleotide DNA (mean size 150 bp). Two libraries were generated from the cDNA fraction a) ~50 ng were used to obtain 5′ gene expression libraries through enzymatic fragmentation, end repair, A-tail, and ligation to adaptors provided in the kit. Unique Illumina sample indexes for each library were introduced through 14 cycles of PCR amplification with primers from the Chromium i7 Multiplex Kit. Library quality was assessed on an Agilent Bioanalyzer 2100, obtaining an average library size of 425 bp) ~ 5 ng of cDNA were used for the generation of T-Cell-Receptor (TCR) VDJ libraries. The cDNA was first enriched for full-length (TCR) VDJ regions by nested PCR amplification with specific VDJ outer and inner primer pairs. The quality and quantity of the VDJ region enrichment were assessed using an Agilent Bioanalyzer 2100 (Santa Clara, CA). Libraries were made as for 5′ gene expression except that 9 PCR amplification cycles were used. The average TCR library size was 542 bp (10x Genomics, PN-1000071). The library for the dextramer oligonucleotide DNA was completed by introducing an Illumina sample index through a 9 cycle PCR amplification step. Then, 5′ gene expression libraries and dextramer oligonucleotide libraries were pooled at a 10:1 ratio and clustered on an Illumina HiSeq4000 on a paired-end flow cell and sequenced for 28 cycles on R1 (10x barcode and the UMIs), followed by 8 cycles of i7 Index (library index), and 98 bases of R2 (transcript or oligonucleotide), obtaining about 120 M clusters per sample. TCR libraries were clustered on a paired-end flow cell and sequenced for 150 cycles, followed by 8 cycles of I7 index (library index), obtaining about 20 M clusters per sample. Primary processing of sequencing images was done using Illumina's Real Time Analysis software (RTA) v3.4.4. 10x Genomics Cell Ranger Single Cell Software suite v3.0.2 was used to perform sample demultiplexing, alignment (mm10), filtering, UMI counting, single-cell 5′end gene counting and associated feature barcoding of AH1 Dextramer using barcode sequence CAAGCC-GACTGCTCC, TCR assembly, annotation of paired VDJ and performing quality control. For each sample, data from approximately 4300 single cells passed quality control and were sequenced to about 50% saturation.

Downstream single-cell analysis was performed using *Seurat* v3.0.2[20] in R v3.5.2[79]. Briefly, for each condition, a Seurat object was created from the 5′ gene expression and AH1-dextramer (feature barcoding) data, and the VDJ information was added as metadata. After pre-processing of the data according to the Seurat workflow, only Cd3+Cd4+Cd8- and Cd3+Cd4-Cd8+ cells (as determined by transcriptional levels) with >200 and <5000 features and <5% mitochondrial gene content were included for downstream analysis. Then, datasets corresponding to the different conditions were integrated using functions *FindIntegrationAnchors* and *IntegrateData* in *Seurat*[20]. To reduce bias, 1920 cells were randomly selected from each condition prior analysis (original numbers 1920, 2317, 3363 and 2447 cells for untreated, CTLA4, RT, and RT + CTLA4 groups, respectively). Then, the dimensionality was reduced by calculating principal components and eventually reduced to 2 dimensions using the *Uniform Manifold Approximation and Projection* (UMAP) algorithm[21]. Finally, clustering of cells was performed. Furthermore, the functional state of each single cell was estimated using the ProjecTILs computational method by applying the "tumor-infiltrating T lymphocytes (TIL) atlas" as a reference dataset (16,803 single-cell transcriptomes of TILs from 25 B16 melanoma and MC38 colon adenocarcinoma tumors curated from 6 studies)[22]. Differentially expressed genes were calculated between all Seurat clusters and/or ProjecTILs functional states.

Differential expression of genes was considered statistically significant if the log2 fold change > 1, the adjusted *p*-value > 0.01, and the percent in the target single cell population > 60%. Enrichment of functional states or clusters was determined using Fisher Exact test on a 2 × 2 contingency table; *p*-values were adjusted for multiple comparison using False Discovery Rate method using the function *p.adjust*

in R. Enrichment was considered statistically significant if *p* < 0.05 and odds ratio > 1.5 or < −1.5.

**Enrichment of single cell clusters in public gene expression datasets.** The publicly available breast cancer METABRIC[39] and TCGA skin cutaneous melanoma (SKCM) datasets were used to determine the association between transcriptional activity of the different T cells clusters defined in the single-cell experiments and patient survival. Both datasets were downloaded from http://www.cbioportal.org and loaded into R v3.5.2, which was used for all subsequent analyses. For the METABRIC dataset, only data from patients with triple-negative breast cancers (patients with "ER_STATUS", "PR_STATUS" and "HER2_STATUS" set to "-". *n* = 499) was used. Single-cell cluster-specific enrichment scores was calculated for each sample using genes that i) was upregulated > $2^{0.5}$ fold change with adjusted *p*-value < 0.01 in the single-cell experiments and ii) had > 1.3 fold change with adjusted *p*-value < 0.01 increased gene expression levels in 4T1 tumors following RT+CTLA4i blockade combination treatment as measured by bulk RNA-seq (see method "Bulk tumor RNA sequencing"). For each patient and cluster, an enrichment score (ES) was calculated as:

$$ES = \frac{\sum_{i=1}^{n} GE_i \times \log_2 FC_i}{n} \tag{5}$$

where *i* is the gene, *n* the number of genes in signature, *GE* the gene expression of gene$_i$ for each patient, and *FC* the fold change of gene$_i$ within each single cell cluster. As shown in the equation, single cell cluster-specific gene expression $\log_2$ fold changes were used as weighting factors prior to averaging the expression of genes. *Survival* v2.44.1.1 and *survminer* v0.4.6 were used to construct univariable and multivariable (METABRIC: age, menopausal state, Nottingham Prognostic Index, and scaled CD3E gene expression; SKCM: age, AJCC pathologic tumor stage, and scaled CD3E gene expression) Cox proportional hazard models from scaled enrichment scores and log-rank survival models using scores as ordinal data (50% percentile; low and high). For the SKCM dataset, all patients with complete TNM data were included (*n* = 417). AJCC pathologic tumor stage was transformed to numeric values prior to analysis (stage I, IA, IB: 1; stage II, IIA, IIB, IIC: 2; stage III, IIIA, IIIB, IIIC: 3; stage IV: 4). For all analysis, patients alive or who died of other causes were censored.

**Assessment of the anti-CD40 activated cDC1s gene signature in mouse tumor bulk RNA sequencing data.** A gene signature of anti-CD40 activated cDC1s has been previously defined in MC38 mouse tumors[52]. The expression of the genes that define this signature (Relb, Etv3, Batf3, Aebp2, Nfkb2, Ccl22, Ccl5, Il15, Ccr7, Il15ra, Plxnc1, Pmp, Cd40, Birc2, Fscn1, Anxa3, Cacnb3, Nudt17, Socs2, Tspan3, Serpinb6b) was investigated to assess the impact of CD40 on the tumor microenvironment. Volcano plots of the genes in this DC1 gene signature were generated using the package *EnhancedVolcano* package in R. The average expression (generated via DESeq2) of the genes was determined as the arithmetic mean of the log scale gene expression data.

**AH1-specific TCR repertoire.** Using the TCRB CDR3 sequences of AH1-specific CD8+ T cells previously published[9] and from the 10X single cell VDJ experiments from this study, we constructed a signature comprised of the TCRB sequences of T-cells that are reactive towards the AH1-antigen. Briefly, for the previously published AH1-reactive TCRs: AH1/H2-Ld-pentamer+ CD8+ T cells from individual 4T1 tumors of untreated or RT+CTLA4i-treated mice (*n* = 5 mice/group) were sorted using a BD FACSAria-II (BD Biosciences). Then, DNA was isolated from the sorted cells and submitted to Adaptive Biotechnology for TCRB CDR3 sequencing[9]. The AH1-specific TCR repertoire can be found in Supplementary Data File 1.

## Flow cytometry analysis of tumors and lungs

Lungs were perfused by injection of 10 mL cold PBS through the right ventricle prior to collection. Tumors and lungs were excised, chopped into small pieces, and enzymatically dissociated using Mouse Tumor Dissociation Kit (Miltenyi Biotec, cat #130-096-730) and Lung Dissociation Kit (Miltenyi Biotec, cat #130-095-927) on a gentleMACS Octo Dissociator (Miltenyi Biotec). Tissue homogenates were resuspended in 10% FBS-RPMI-1640 (Corning) and filtered on a 70 μm strainer to remove large debris. Cells were washed in cold PBS. Lung cell suspension was resuspended in a working solution of red blood cell (RBC) Lysis Buffer (eBioscience, cat #00-4300-54) and incubated for 2 min at room temperature. Cells were washed twice in PBS. Cells stained with AH1-dextramer PE (Immudex, cat #JG3294-PE) were pretreated with 50 nM dasatinib (Sigma-Aldrich) for 30 min at 37 °C, before the addition of the PE-conjugated AH1-dextramer were added per test and cells were incubated on ice for 30 min and washed in PBS. Viability dye staining (Zombie Aqua, Biolegend cat #423101 or Zombie UV, Biolegend cat #423107) was performed prior to surface staining with the following antibodies: Tumors: Panel 1: CD3 (cl17-A2; BUV395), CD4 (clRM4-4; BUV496), CD8 (cl53-6.7;BUV615), CD69 (clH1.2F3; BUV737), CD44 (clIM7; BUV805), GITR (clDTA-1; super bright 436), CD19 (cl 6D5; BV510), Epcam (clG8.8; BV510), CD11b (clM1/70; BV510), CD11c (clN418; BV510), CD62L (cl MEL-14; BV650), PD1 (cl 29 F.1A12; BV785), CD25 (cl PC61; PE fire 640), CD45 (cl 30-F11; alexa fluor 532), TIM3 (cl 5D12/TIM3; BB700), CD40Lg (cl SA047C3; PE), OX40 (cl OX-86; PE-Cy7), CTLA4 (cl UC10-4F10-11; APC-R-700), LAG3 (cl C9B7W; APC eFluor 780), TIGIT (cl GIGD7; PerCP eFluor 710) Panel 2: CD3 (cl 145-2C11; BUV395), CD45 (cl 30-F11; alexa fluor 532), Epcam (clG8.8; BV510), CD11c (cl N418; BUV737), CD11b (cl M1/70; pacific blue), F4/80 (cl BM8; BV421), Ly6c (cl HK1.4; BV711), Ly6g (cl 1A8; PE fire 640), CD40 (cl 3/23; PE), MHC II (cl M5/114.15.2; FITC), CD80 (cl 16-10A1; PE Cy7), CD86 (cl GL1; APC Cy7), CD206 (cl MR6F3; PerCP eFluor 710), XCR1 (cl ZET; APC). Lungs: CD3 (cl17-A2; BUV395), CD45 (cl 30-F11; alexa fluor 532), CD4 (cl GK1.5; PerCP-Cy5.5), CD8 (cl53-6.7;BUV615), CD25 (cl PC61; PE fire 640), CD69 (cl H1.2F3; BV421), PD1 (cl 29F.1A12; BV785), CXCR3 (cl S18001A; BV711), CD40 (cl 3/23; PE Dazzle 594), CD11c (cl N418; BV510), CD11b (cl M1/70; eFluor 450), XCR1 (cl ZET; BV650), F4/80 (cl t45-2342; BUV395), MHC II (cl M5/114.15.2; BUV 737), CD80 (cl 16-10A1; PE-Cy7). Surface staining was done in Brilliant Stain Buffer (Thermo Fisher Scientific). Cells were fixed and permeabilized on ice using the Foxp3 / Transcription Factor Staining Buffer Set (eBioscience, cat #00-5523-00). Intra-cellular staining with Ki67 (cl 16A8; PE Dazzle 594) and/or FOXP3 (cl MF23; alexa fluor 647) was performed on ice.

The list and specifications of antibodies used for each panel are detailed in Supplementary Tables 3, 4, and 5. CountBright absolute counting beads (Thermo Fisher Scientific) were added to the tumor samples before acquisition.

All three panels were optimized for use on a Cytek Aurora spectral flow cytometry platform with a 5 lasers (16UV-16V-14B-10YG-8R) fixed configuration.

Data were analyzed using FlowJo v10 (Supplementary Figs. 24, 25) software, DownSample (v3.3.1) and FlowSOM (v3.0.18) plugins and cloud-based analysis platform OMIQ (https://omiq.ai).

High dimensional analysis was conducted on PD1[+] antigen-experienced (CD44[+] CD62L[-]) CD8[+] T cells. Data were downsampled to 1000 T cells per sample from the RT + CTLA4i group; 8 out of 10 samples met the 1000 cells criteria and were concatenated for the rest of the analysis. Opt-sne was run using standard imputs (perplexity = 30, iterations = 1000) based on 4 channels (PD1, Lag3, TIGIT, Tim3). FlowSOM-based metaclustering was then performed (Metaclusters = 6/5/4/3, clusters = 100, training iterations =10). Elbow metaclustering analysis identified 6 metaclusters based on the 4 biomarkers selected. Analysis of the expression of LAG3, TIGIT, TIM3 on these 6 metaclusters to define their activation and/or exhaustion state showed

redundancy between the clusters. Metaclustering was thus ran imposing the definition of 5, 4 or 3 distinct metaclusters. The most relevant separation of the PD1[+] population was achieved using 4 metaclusters, that are overlayed on the opt-sne visualization of the PD1[+] CD44[+] CD62L[-] CD8[+] population as a color dimension (Fig. 5d). Expression of PD1, TIM3, TIGIT, LAG3 and Ki67 on cells from each metaclusters are detailed on Supplementary Fig. 15c, d.

## Lung metastasis quantification

Lungs were collected and processed as described above. Single cell suspension was resuspended in medium containing 60 μM 6-thioguanine (Sigma-Aldrich), serially diluted in 6-well tissue culture plates[80] and incubated at 37 °C, 5% CO2. After 11 days, plates were fixed in ethanol and colonies stained in 0.5% crystal violet (Sigma Aldrich) diluted in 25% methanol and counted. Accounting for the dilution factors, data are expressed as the number of lung metastatic colonies per mouse.

**Statistical analyses.** For the survival surgery experiments, statistical significance in tumor volume growth between groups was determined with 2-way repeated measures ANOVA between day 15–21 and $t$ test at day 21. Tukey's and Holm's method for adjusting p-values corrected for multiple comparison was used for the ANOVA and $t$ tests, respectively. For the experiments assessing the therapeutic effect of adding antibodies against PD1, LAG3, CD40, GITR, and OX40 to the RT+CTLA4i therapy, statistical significance of the differences in tumor growth rates between the treatment groups was calculated in R (version 3.6.1) using packages nlme and multcomp (versions 3.1-140 and 1.4-10, respectively). Briefly, longitudinal tumor sizes (tumor volume data was square rooted prior to analysis) were modeled using a linear mixed-effects model. To ensure the data was fitted properly, the model parameters and residuals were assessed. P-values reported were calculated using a general linear hypothesis testing method for pairwise differences in tumor growth between any two treatments, and then adjusted for multiple comparison using the single-step method. Flow cytometry data were analyzed by Kruskal-Wallis and post-hoc Dunn's test using Prism v9.4.1. Changes in log-transformed tumor volume were compared between groups by Mann Whitney tests on Log-transformed values using Prism v9.4.1. Linear mixed-effects regression models were used to estimate the number of lung metastases in each treatment group while accounting for potential within mouse correlations. Generalized linear hypothesis testing was used to evaluate contrasts of interest (Control / RT CTLA4 and RT CTLA4 CD40 *VS* all the other groups) while control for potential experiment-specific effects. Square root transformed data were used to ensure the underlying model assumptions were satisfied. *P*-values were adjusted for multiple comparisons by controlling the false discovery rate. R (version 4.2.3)

## Reporting summary

Further information on research design is available in the Nature Portfolio Reporting Summary linked to this article.

# Data availability

Raw RNA sequencing data generated during the current study have been deposited in NCBI's Sequence Read Archive (SRA) database under the accession number of PRJNA596248 and sample annotation is available in Supplementary Data 4 for RNAseq, and Supplementary Data 5 for TCRseq. CDR3B sequences of AH1-specific TCRs are listed in Supplementary Data 1. The raw DNA TCR sequence data have been deposited into the ImmuneACCESS project repository of the Adaptive Biotechnology database [https://doi.org/10.21417/NR2023NC], and sample annotation is available in Supplementary Data 6 and 7. Transcriptome profiling data available for breast cancer patients were downloaded from the publicly available METABRIC and The Cancer

Genome Atlas (TCGA) databases from the http://www.cbioportal.org portal along with corresponding clinical information. The remaining data presented in the manuscript are available within the Article, Supplementary Information, Source Data file and Supplementary Data 2 and 3. Source data are provided with this paper.

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

## Acknowledgements

This work was supported by NIH (R01CA198533 and R01CA201246), to S.D., and BCRF-23-053 to S.D. and S.C.F. We acknowledge the use of the Genomics, the Epigenomics, the CLC Flow Cytometry and the Human Immune Monitoring Core facilities of Weill Cornell Medicine. We also acknowledge the use of the Integrated Genomics Operation Core, funded by the NCI Cancer Center Support Grant (CCSG, P30 CA08748), Cycle for Survival, and the Marie-Josée and Henry R. Kravis Center for Molecular Oncology at the Memorial Sloan Kettering Cancer Center. In addition, we acknowledge the use of data generated by the TCGA Research Network: https://www.cancer.gov/tcga. We thank Dr. Roberta Zappasodi for helpful suggestions about the use of anti-GITR antibodies.

## Author contributions

N.-P.R and S.D. conceived the study and wrote the manuscript. N.-P.R. designed and performed the sequencing experiments and data analysis. M.C. designed and performed the tumor growth and treatment experiments and the analysis of the immune infiltrate. C.L. helped develop survival surgery experiments. C.L., E.W. and S.S. contributed to tumor

growth and treatment experiments and aided in data analysis. C.S. and A.A. helped with the 10X-single cell sequencing. J.S. helped with the TCR sequencing and analysis. X.K.Z. supervised statistical analysis. T.Z. helped with RNA sequencing analysis. S.C.F. edited the manuscript. All authors approved the final version of the manuscript.

## Competing interests

The authors declare that they have no competing interests related to this work. However, S.D. has received compensation for consultant/advisory services from Lytix Biopharma, EMD Serono, Ono Pharmaceutical, Genentech, and Johnson & Johnson Enterprise Innovation Inc., and research support from Lytix Biopharma, Nanobiotix, and Boehringer-Ingelheim for unrelated projects. S.C.F. is/has been holding research contracts with Merck, Varian, Bristol Myers Squibb, Celldex, Regeneron, Eisai, and Eli-Lilly, and has received consulting/advisory honoraria from Bayer, Bristol Myers Squibb, Varian, Elekta, Regeneron, Eisai, AstraZeneca, MedImmune, Merck US, EMD Serono, Accuray, Boehringer Ingelheim, Roche, Genentech, AstraZeneca, View Ray, and Nanobiotix.
