## [Peer Review File · Nature Communications]

Immunotherapy targeting different immune compartments in combination with radiation therapy induces regression of resistant tumorsREVIEWER COMMENTS

Reviewer #1 (Remarks to the Author): with expertise in cancer, radio-immunotherapy

This manuscript is well written and the authors have comprehensively profiled T cell subsets within the TME of the commonly used 4T1 triple negative breast cancer model. The authors have identified several intriguing and hypothesis generating findings. They also have used sound methods, statistics, and data analysis.

However despite many strengths, there are some additional studies that could improve the impact of this manuscript.

Some more mechanistic studies knocking out or depleting some of identified necessary T cell cluster subsets would have been helpful.

The authors importantly mention the potential of the CTLA4 clone they used which can deplete Tregs in the discussion. There are CTLA4 clones that are readily available that do not deplete Tregs and it may be worth testing if this activity is necessary for their demonstrated treatment effect.

In figure 6, it would be helpful if the authors include tumor growth and survival curves as they did for the AT3 model. Also having abscopal data in both models (add this data for 4T1) would be helpful if the authors wish to show that an abscopal response is an important part of their treatment effect.

In figure 7, inclusion of RT only, and dual checkpoint only controls would be helpful. It is unclear if RT is needed to help delay metastases or if it is primarily due to the systemic therapy.

For 4T1 and AT3 efficacy studies both studies are well powered, however a second replicate to show reproducibility would improve rigor.

While the authors have nicely performed a comprehensive analysis of TILs with RT with and

without anti-CTLA4, the mechanistic link for adding anti-CD40 is not well described or studied. The authors have shown that adding anti-CD40 to anti-CTLA4 and RT improves efficacy as others have also shown. However as much of the initial manuscript has focused on characterization of TILs without in depth examination of innate immune cells, it seems a bit disjointed with the latter part of the manuscript where efficacy studies show importance of CD40 which has been shown to activate innate immune cells. The connection between the first and second parts of the manuscript seems quite loose and generating a more direct mechanistic link would be helpful.

Reviewer #2 (Remarks to the Author): with expertise in breast cancer, immunotherapy
The manuscript from Rudqvist et al is very thorough and describes the benefit of treating 4T1 TNBC tumors with single agent or combination of anti-CTLA4 and radiation therapy (RT). Careful analysis of the T cell populations recruited into the tumor are provided, including scRNAseq analysis of these T cells. The authors convincingly demonstrate that RT + anti-CTLA4 therapy inhibits tumor growth, but addition of anti-CD40 induces tumor regression and enhanced survival on mice bearing 4T1 tumors. In contrast, addition of anti-PD1, LAG3, TIM3, TIGIT, OX40 or GITR did not improve the response to CTLA4+RT. RT + a-CTLA4 therapy did not increase the % GITR+ CD4 T cells, or those that were Ox40+. However agonist CD40 antibody induced regression of 7/9 4T1 tumors treated with RT + CTLA4Ab. This combination resulted in a reduction in metastasis and an increase in CD69+CD8+ T cells and increased CD69+/CXCR3+ CD4 T cells. Addition of zCD40 also increased the %CD40+ macrophages, the % CD11c+F4/80 cells, and decreased the % CD40+/CD11c+ F4/80 neg DCs while increasing the MFI of that population of DCs.

The authors show that the anti-CTLA4 treatment increases the number of CD4+ T helper cells while RT enriches for CD8+ T cells with effector memory, early activation and exhausted phenotypes. Of interest, CD40L was upregulated in the CD4Th1 population of cells, indicating a potential for enhanced recruitment and activation of DC in response to a-CTLA4.

Gene expression signature data from the mouse model are compared with gene signatures

of functional states of T cells in TNBC patients from the METABRIC dataset and from the TCGA SKCM data set.

The data in the 4T1 model are reproduced to some extent with the AT3 TNBC model-- though regression was not observed with the triple combination therapy as was seen with the 4T1 cells.

The methodology is sound and meets expected standards. Statistical analyses and methods are appropriately detailed.

While the manuscript is quite strong, there are a few things that require additional explanation:

1) The t-test used in Figure 1B shows significant differences between RT and RT+CTLA4Ab. However, the SD overlaps between these two growth curves and the data appear non-significant. Was a one-sided or 2-sided t test used and justification for choosing this specific t test needs explanation.

2) there are no data showing toxicity for the combined anti-CTLA4, RT and agonist anti-CD40 treatment. The authors chose a very high concentration of CD40Ab--100 micrograms as compared to the preferred 30 micrograms. This higher dose could introduce higher toxicity and non-specific inflammatory responses.

3) The authors infer that CD40+DCs are the target for the anti-CD40 therapy, but this needs to be validated by using either depletion of CD40+DCs or using a CD40 KO mouse.

4) It is important to show how T cell clonality is impacted by addition of agonist CD40 Ab to the RT + a-CTLA4 regime.

Reviewer #3 (Remarks to the Author): with expertise in cancer, radio-immunotherapy

The manuscript "Immunotherapy targeting different immune compartments in combination with radiation therapy induces regression of resistant tumors." By Rudqvist, et al. describes changes in T cells activation after Radiation therapy (RT), immune checkpoint inhibition (ICI) by antibody-based CTLA-4 inhibition (CTLA-4i) and combination treatment in the 4T1 triple negative breast cancer model (TNBC). Subsequently they also conclude that adding additional ICI does not increase the RT+CTLA4 response rate.

Unfortunately, at this stage the possible findings and conclusions are difficult to interpret due to

- a concern of rigor and reproducibility. Namely, the first 4 figures all depict genomic data from a single animal experiment (n=4-6 per group).
- Tumor growth curves of the individual mice are needed. Some of the data shown might be strongly influenced by an outlier, esp. when n=4.
- Proteomic verification is missing for several conclusions.
- Many reports show that 4T1 is not poorly immunogenic as the authors state.
- It is unclear why the authors compared some of the possible findings in 4T1 model with publicly available melanoma patients data.
- The Rt regime is not clinical relevant, and thus it is difficult to interpret what the data of a specific mouse model and used intervention extrapolates to the clinic.
- The limitations of some of the functional immunological readouts have not been addressed.
- Essential controls are missing, e.g. Fig.7.
- The addition of CD40 treatment to RT + CTLA4 increased the percentage of CD8+CD69+ by 5 fold, yet this did not reduce 4T1 colony formation. This is not addressed and goes against the overall premise of the study.
- Several reports are out related to the 4T1 immuno phenotype, albeit without radiation, these need to be incorporated in the discussion.

Very minor: although generally well written it has some a few typos / inconsistencies.

Reviewer #4 (Remarks to the Author): with expertise in cancer, TCR-repertoire

General comments

In this study, the authors, studied in depth the effect of combined radiotherapy and CTLA4 inhibitor on the immune response in mice. The manuscript is well written although some parts are really dense. They combined in a first part of the study TCR sequencing, bulk and single RNAseq as well single cell TCRseq and flow-cytometry. This first part of the study is quite straightforward although I have some concerns regarding the TCR data generation quality and the analyses as well as on the single cell analysis comparison of projectTILS versus Seurat Cluster (see detailed comments). The second part which is more flow cytometry based on a different tumor mouse model with an additional treatment added to the two first lack unfortunately a direct link with the first part to identify what the third treatment, anti-CD40, bring molecularly speaking. If this part should stay, at least RNAseq and eventually TCRseq should be added. Finally, details on the sequencing data (bulk TCR, bulk RNA and single-cell) should be added as supplemental for quality check.

Major comments

The method used for bulk TCR sequencing is far from being the most reproducible as shown by previous method benchmarking (Rosati 2016, Liu 2018, Barennes 2020). ImmunoSEQ was also mentioned as being applied on some instances. When, on what? Authors should clearly specify the number of sequences obtained, the number of clonotypes at the nucleotide (nt) and

amino acid (aa) level, the percentage of aligned reads.... Also, the R packages used for the analysis should be mentioned and codes published as a git if in-house developed.

Shannon clonality as it is calculated is not a clonality assessment, rather it seems to be the inverse of the normalized Shannon entropy, therefore not assessing clonality but a measure of the diversity, where the higher the measure the lower the diversity. This could be extrapolated as an increased clonality associated with clonal expansions, but need to be demonstrated. Authors should use the Renyi profile (or Hill entropy) for instance, in order to show how much expanded clonotypes are more prevalent in the post treatment.

A complementary possibility is to simply calculate the normalized cumulative frequency of

the top X% (or top n sequence if the size of the repertoires are comparable). In Figure S2, the cumulative frequency of ranked rearrangements goes from 1 to 100000. Are the number of rearrangements comparable between samples? If not, a normalization is required.

Figure 1D shows the JSD at the VJ combination level between paired pre and post treated animals. It would be interesting to determine the similarity between the top clonotypes (including the CDR3 region) post-treatment with the whole pre-treatment repertoire in order to determine whether the post-treatment expansions result from intra-tumor expansions (including rare clonotypes that may not be captured in the top 700 from the pre-treated group) or from newly recruited cells. For this, a Jaccard and Morisita horn similarity measures might help.

Statistics used in the manuscript should be more detailed. In figure 1 and S3, parametric tests have been used for an n=5 max and without normality assessment. First check for normality and in all the case, given the small sample size, non-parametric test should be favored, median instead of mean should be represented and correction for multiple testing included.

In Figure S3, when post-treatment between conditions are compared, the author should also indicate the absence of pre-treatment Shannon « clonality » between groups of mice.

AH1 specific-TCR signature needs to be detailed: how many unique TCRs, how related between each other, what about their distribution in the new data sets.

The AH1-specific TCR were inferred based on CDR3 sequences obtained from purified CD8 T-cells. Since the experiments from which they identified the AH1 specific TCRs and the new ones are done in congenic mice with the same genetic background, authors should look at the entire rearrangement (including V and J). Using previously published data on AH1 CD8 T cells is not sufficient to ensure that the CDR3s they are identifying in the tumor are from AH1 CD8 T cells.

In Figure 2 and related text, authors should not state that the increase in CD3, CD8 and CD4 correspond to an increased infiltration as an active process, unless chemokine receptors are

also observed. It could also be a proliferation due to decreased blockade of anti-inflammatory cells or molecules due to the treatment.

In figure 3, authors subsampled large datasets to the lowest in order to avoid bias due to sample size. Could they detail in supplementary how many cells per condition were in fact obtained in order to appreciate the magnitude of subsampling from the larger dataset. If relevant, multiple subsamples of the largest dataset could be generated and similarity evaluate to ensure the representativeness of the subsample.

Figure 3 B is surprising as it is well known that CD4 expression is usually low from single-cell data. Are the Expression levels per panel normalized per gene analyzed? If so, authors should mention it.

Then, they looked at TILs signature to deconvolute T-cell subsets and identified different cell subset. The subdivision of those ProjectTILs clusters back to the Seurat cluster is a bit more concerning. Indeed, the heatmap of the ProjectTILs signature shown in supplementary looks like there is within the Seurat clusters a high heterogeneity. Is that a biological « reality » or the results of heterogeneous gene capture? Are all the Seurat clusters consistent in terms of gene counts/UMI?

Finally, although I found the last part about combination of anti-CD40 together with RT and CTLA4i interesting (Figure 5 to 8), it lacks the depth and precision of the analyses shown till figure 4. Therefore, even if some link could be made between the two studies through the flow cytometry data, we cannot really what anti-CD40 is doing at the molecular level, and conclude on possible synergistic effects of the combo therapy. This part could be left for another story where some single-cell or bulk transcriptome analyses could be performed.

Data

- It is not clear if the AH1-TCR data are from new experiments or a mixture of previous experiments (Rudqvist et al, 2018). Please clarify in the method (including the description of the signature).
- Newly generated TCR data should be shared via NCBI. The PRJNA596248 Bioproject does not include such data. Raw data must be obtained from iRepertoire. For TCR data, please use the template designed by the AIRR Community, which reports required metadata for

AIRR-seq data sharing and reuse. If really not possible, authors could share the data through federated databases, such as iReceptor (<http://ireceptor.irmacs.sfu.ca/>) starting from tsv files. The PRJNA project indicated in the manuscript seems to target previously obtained data.

Minor comments

Several typos to be corrected across the manuscript

RE: NCOMMS-22-43189

Immunotherapy targeting different immune compartments in combination with radiation therapy induces regression of resistant tumors

We would like to thank the reviewers for the valuable and constructive criticism of our manuscript.

Main changes: We have performed additional experiments addressing the mechanisms of anti-CD40-mediated immune effects in mice treated with RT+CTLA4i, and results are shown in the new Figures 7 and 8 of the revised manuscript. As detailed below, we have also provided additional data in Supplementary materials and made additional changes to the manuscript to address the reviewer's comments. Former Figure 7 has been replaced by Figure 9. Changes to the text are marked in red ink in the tracked version of the revised manuscript.

Reviewer #1

This manuscript is well-written and the authors have comprehensively profiled T cell subsets within the TME of the commonly used 4T1 triple negative breast cancer model. The authors have identified several intriguing and hypothesis generating findings. They also have used sound methods, statistics, and data analysis. However despite many strengths, there are some additional studies that could improve the impact of this manuscript.

1) Some more mechanistic studies knocking out or depleting some of identified necessary T cell cluster subsets would have been helpful.

Authors' response: We have considered the possibility of using genetic or pharmacologic tools to directly test the role of specific T cell clusters but feel that the latter represent dynamic T cell functional states and it will not be possible to selectively affect one subset without perturbing others.

2) The authors importantly mention the potential of the CTLA4 clone they used which can deplete Tregs in the discussion. There are CTLA4 clones that are readily available that do not deplete Tregs and it may be worth testing if this activity is necessary for their demonstrated treatment effect.

Authors' response: We agree with the reviewer that the role of Tregs in the response to RT+anti-CTLA4 has not been directly addressed in our study. The aim of our study was to improve tumor response and we found that additional treatment with antibodies to OX40 and GITR that have been shown to effectively target Tregs did not have an effect, while anti-CD40 did. Thus, we focused on myeloid cells in this study, and plan to investigate Tregs in future studies. We have added a sentence in the Discussion section (page 22, lines 3-6) to acknowledge that our study did not directly investigate the role of Tregs.

3) In figure 6, it would be helpful if the authors include tumor growth and survival curves as they did for the AT3 model. Also having abscopal data in both models (add this data for 4T1) would be helpful if the authors wish to show that an abscopal response is an important part of their treatment effect.

Authors' response: The tumor growth curves for the experiments presented in Figure 6 are provided in Supplementary figure S17. Long-term survival was seen only in mice with MCR, which were rechallenged with 4T1 cells, as shown in Figure S17C.

In the 4T1 model, tumor growth outside of the irradiated “primary” tumor occurs spontaneously in the lungs which are seeded with metastatic cells starting a week after the initial subcutaneous injection of 4T1 cells. Due to the high metastatic proficiency of 4T1 cells the presence of multiple subcutaneous tumors accelerates the lung metastases and reduces the window of opportunity to measure tumor response in a second subcutaneous tumor. Thus, we measured the systemic (abscopal) effect of treatment in the lungs.

5) In figure 7, inclusion of RT only, and dual checkpoint only controls would be helpful. It is unclear if RT is needed to help delay metastases or if it is primarily due to the systemic therapy.

Authors’ response: We agree with the reviewer’s point and performed a new experiment including all the treatment groups. These data are shown in new Figure 9 (which replaces the prior Figure 7). Anti-CTLA4+anti-CD40 reduced lung metastases, and although the triple combination group showed a more significant reduction compared to control, the difference between the combination of systemic therapies used with and w/o RT was not statistically significant. We have previously shown that RT is required to achieve control of lung metastases in anti-CTLA4 treated 4T1 tumor-bearing mice (PMID 15701862 and 19147765). The current results suggest that RT may be dispensable for lung metastasis control when anti-CTLA4 is used with anti-CD40. Analysis of lung T cells showed that CTLA4i increased tumor-specific CD8 T cells, while anti-CD40 increased T cell activation. However, we cannot exclude a contribution of non-T cells to the anti-metastatic effect of anti-CD40. We plan to address these questions in future studies. As mentioned above, the aim of our study was to identify a combination therapy to overcome tumor resistance to RT+CTLA4i: whereas the addition of anti-CD40 dramatically improved the rejection of the irradiated tumor it did not further improve the inhibition of lung metastases obtained by treatment with RT+CTLA4i in 4T1 bearing mice. In contrast, addition of anti-CD40 was required to achieve control of a non-irradiated subcutaneous tumor in AT3 bearing mice treated with RT+CTLA4i. These results suggest that CD40 agonism may be required for the effectiveness of RT+CTLA4i in established tumors that are dominated by an immune suppressive myeloid infiltrate such as 4T1 and AT3. However, in the settings of lung micro-metastases all combination therapies were beneficial, but none was truly superior. We have added a paragraph (page 24, lines 22-30, and 25, lines 1-2) discussing the potential clinical significance of these findings in the Discussion section.

6) For 4T1 and AT3 efficacy studies both studies are well powered, however a second replicate to show reproducibility would improve rigor.

Authors’ response: The reproducibility of the tumor responses to the combination of RT+CTLA4i+anti-CD40 was shown in four independent experiments for 4T1 (Figure 6B and 6E, and Supplementary figures S18 and S21), and two experiments for AT3 (Figure 10B and 10E). Each experiment was designed to provide additional information, in accordance with the three Rs (replacement, reduction, refinement) principle of humane use of vertebrate animals in research to maximize the information obtained per animal.

7) While the authors have nicely performed a comprehensive analysis of TILs with RT with and without anti-CTLA4, the mechanistic link for adding anti-CD40 is not well described or studied. The authors have shown that adding anti-CD40 to anti-CTLA4 and RT improves efficacy as others have also shown. However as much of the initial manuscript has focused on characterization of TILs without in depth examination of innate immune cells, it seems a bit disjointed with the latter part of the manuscript where efficacy studies show importance of CD40 which has been shown to activate innate immune cells. The connection between the first and

second parts of the manuscript seems quite loose and generating a more direct mechanistic link would be helpful.

Authors' response: We thank the reviewer for this suggestion, and have performed new experiments, shown in new Figures 7 and 8, to investigate the effects of anti-CD40 on the myeloid infiltrate in tumors. Briefly, we found a significant enrichment – compared to control - of a gene signature of anti-CD40 activated cDC1s only in tumors treated with RT+CTLA4i+anti-CD40 (Figure 7A and B). Consistently, flow cytometry analysis of the myeloid tumor infiltrate showed increased representation of XCR1+ cDC1 expressing higher levels of costimulatory molecules in mice treated with anti-CD40 compared to the respective treatment groups without anti-CD40. A corresponding increase in frequency and activation of XCR1+ cDC1 was also seen in the draining lymph node (Figure 8). Anti-CD40 also increased significantly AH1-specific TCR repertoire sharing between the tumor and draining lymph node of each mouse as compared to controls (Figure 7C), supporting the interpretation that the main effect of anti-CD40 is to increase cDC1 activation and tumor-specific T cell priming.

Reviewer #2

The manuscript from Rudqvist et al is very thorough and describes the benefit of treating 4T1 TNBC tumors with single agent or combination of anti-CTLA4 and radiation therapy (RT). [...] While the manuscript is quite strong, there are a few things that require additional explanation:

1) The t-test used in Figure 1B shows significant differences between RT and RT+CTLA4Ab. However, the SD overlaps between these two growth curves and the data appear non-significant. Was a one-sided or 2-sided t test used and justification for choosing this specific t test needs explanation.

Authors' response: We have added a new Supplementary Figure (S1) showing the individual mice growth curves in each group and the tumor volume at day 21. P values were calculated using a t-test adjusted for multiple comparisons using the Holm method. Statistical theory does not prohibit overlapping standard deviations as the t-test is testing the null hypothesis that there is no difference in the average between two groups.

2) There are no data showing toxicity for the combined anti-CTLA4, RT and agonist anti-CD40 treatment. The authors chose a very high concentration of CD40Ab--100 micrograms as compared to the preferred 30 micrograms. This higher dose could introduce higher toxicity and non-specific inflammatory responses.

Authors' response: We have chosen a dose of 100ug/ mouse given intra-peritoneally since this dose has been used in several mouse studies for systemic anti-CD40 administration whereas lower doses (e.g., 20-30ug) have been more often used for local (intra-tumoral) delivery. In addition, 100ug dose of anti-CD40 was previously used in combination with radiotherapy without significant reported toxicity (PMID: 29844122). Mice were followed closely by us and the veterinary services of our institution. Mild signs of potential toxicity (e.g., some "ruffled" fur) were observed but resolved 1 to 2 days after CD40-agonist treatment completion.

To assess toxicity mice were weighed at regular intervals during treatment in a new experiment (the endpoint of the experiment was analysis of the immune infiltrate at completion of treatment, thus mice were euthanized at day 22 post tumor injection for lung collection). As shown in the figures below, we observed mild weight loss (<11% of pre-treatment weight) in the groups that received CD40 agonist. In collaboration with the veterinary services, we found this weight loss acceptable and not interfering with animal wellbeing.

Figure legend: 4T1 tumor-bearing mice were treated with RT (day 12, 13, 14), anti-CTLA4 (day 13, 16, 19) and anti-CD40 (day 16, 18, 20) alone or in combination as indicated. (Top) Mice weight change during treatment. Each dot represents one mouse. (Bottom) Mice weight, mean \pm SD. Groups are indicated by color as in the top panel.

3) The authors infer that CD40+DCs are the target for the anti-CD40 therapy, but this needs to be validated by using either depletion of CD40+DCs or using a CD40 KO mouse.

Authors' response: Because CD40 is expressed by multiple immune cells besides DCs (e.g, macrophages and B cells) instead of depletion we used RNA sequencing and flow cytometry to assess the effects of anti-CD40 on the DCs present in the tumor and draining lymph node. Results of these experiments are shown in new Figures 7 and 8. Briefly, we found a significant enrichment – compared to control - of a gene signature of anti-CD40 activated cDC1s only in tumors treated with RT+CTLA4i+anti-CD40 (Figure 7A and B). Consistently, flow cytometry analysis of the myeloid tumor infiltrate showed increased representation of XCR1+ cDC1 expressing higher levels of costimulatory molecules in mice treated with anti-CD40 compared to the respective treatment groups without anti-CD40. A corresponding increase in frequency and activation of XCR1+ cDC1 was also seen in the draining lymph node (Figure 8).

4) It is important to show how T cell clonality is impacted by addition of agonist CD40 Ab to the RT + a-CTLA4 regime.

Authors' response: We thank the reviewer for this suggestion. We performed TCR repertoire analysis in the tumor and paired draining lymph node. We did not detect a statistically significant increase in clonality in the tumor, although the median clonality was higher in mice receiving RT+CTLA4i+anti-CD40 as compared to control and RT+CTLA4i groups (see figure below).

Notably, we found that anti-CD40 increased significantly AH1-specific TCR repertoire sharing between the tumor and draining lymph node of each mouse as compared to controls (new Figure 7C), supporting the interpretation that the main effect of anti-CD40 is to increase cDC1 activation and tumor-specific T cell priming.

Reviewer #3

The manuscript “Immunotherapy targeting different immune compartments in combination with radiation therapy induces regression of resistant tumors.” By Rudqvist, et al. describes changes in T cells activation after Radiation therapy (RT), immune checkpoint inhibition (ICI) by antibody-based CTLA-4 inhibition (CTLA-4i) and combination treatment in the 4T1 triple negative breast cancer model (TNBC). Subsequently they also conclude that adding additional ICI does not increase the RT+CTLA4 response rate.

Unfortunately, at this stage the possible findings and conclusions are difficult to interpret due to:

1) a concern of rigor and reproducibility. Namely, the first 4 figures all depict genomic data from a single animal experiment (n=4-6 per group).

Authors' response: It is not accurate that Figure 1 to 4 represent data from a single animal experiment. Figure 1 shows the TCR repertoire analysis of mice from four treatment groups (n=6/group), while Figure 2 shows bulk RNA-sequencing data of a subset of mice shown in Figure 1. In accordance with the three Rs (replacement, reduction, refinement) principle of humane use of vertebrate animals in research it is good practice to maximize the information obtained per animal, which can be accomplished by using completely independent methods on the same material. The analysis performed in Figure 1 and 2 was hypothesis-generating and we subsequently performed the experiment presented in Figure 3 (single cell sequencing analysis). Figure 4A show an analysis that combined bulk and single cell data obtained from different mice

in different experiments; in addition it shows the analysis of publicly available patient data to assess the clinical relevance of the gene expression signatures identified in mice.

2) Tumor growth curves of the individual mice are needed. Some of the data shown might be strongly influenced by an outlier, esp. when n=4.

Authors' response: The tumor growth curves of individual mice are shown in Supplementary Figures (please see S17, related to Fig 6; S22, related to Figure 10). Individual mouse level data are also presented in other experiments shown only as Supplementary data, e.g., S18 and S21). We have added a new Supplementary Figure (S1) showing the individual mice growth curves related to Figure 1B. No experiment designed to assess tumor growth in response to treatment was performed with n=4 mice. The minimum number used was n=5 for the control group (tumor growth of untreated mice is very predictable) while 9 to 15 animals per group were used in experimental groups, and treatment efficacy was confirmed in more than one experiment (see also response to point#6 or reviewer#1).

3) Proteomic verification is missing for several conclusions.

Authors' response: The main T cell subsets that were identified in the single cell analysis were validated in flow cytometry experiments (Figure 5). In addition, the activation of DCs in anti-CD40 treated tumors detected by RNA sequencing (new Figure 7) was confirmed by flow cytometry (new Figure 8).

4) Many reports show that 4T1 is not poorly immunogenic as the authors state.

Authors' response: We obtained our 4T1 cells from the investigator who originally developed this cell line (Fred Miller) and have maintained a stock of low-passage cells. We also regularly test our cells for absence of mycoplasma, which is known to greatly affect tumor immunogenicity in vivo. In addition, genetic manipulation of tumor cells (especially overexpression of GFP or luciferase) can greatly increase tumor immunogenicity. These are some of the common reasons for difference in behavior of cells used in different laboratories. Our data in this manuscript clearly demonstrate that the 4T1 tumor we have studied fulfills the definition of a poorly immunogenic tumor based on the very low T cell infiltrate (see Figure 2A and 5A) and resistance to immunotherapy (see Figure 6).

5) It is unclear why the authors compared some of the possible findings in 4T1 model with publicly available melanoma patients data.

Authors' response: We first compared our data with publicly available breast cancer RNA-sequencing data and found that the signature we had discovered was associated with improved survival in patients. To assess whether this finding was restricted to TNBC, we also investigated a melanoma dataset, and found an association also in the latter. This is consistent with the idea that an effective anti-tumor T cell response shares characteristics between disease sites, but that resistance to treatment is likely specific to each indication.

5) The Rt regimen is not clinical relevant, and thus it is difficult to interpret what the data of a specific mouse model and used intervention extrapolates to the clinic.

Authors' response: We beg to disagree with the reviewer, as this RT regimen (8GyX3) or a similar one has been used in several clinical trials testing combinations of RT with immune checkpoint inhibitors in multiple tumor types, including TNBC. For some examples, please see

published papers (PMID: 34015311; 30397353; 31294749) published meeting abstract on TNBC (<https://doi.org/10.1158/1538-7445.SABCS21-PD10-01>), and a review article reporting on the virtual consensus discussion about radiation doses used in the context of cancer immunotherapy by a panel of radiation oncologists and immunologists (PMID: 33827904).

6) The limitations of some of the functional immunological readouts have not been addressed.

Authors' response: We are aware that all assays and endpoints have limitations but this comment is not specific and thus it is not addressable.

7) Essential controls are missing, e.g. Fig.7.

Authors' response: This figure has been replaced with Figure 9, a new experiment that includes all control groups.

8) The addition of CD40 treatment to RT + CTLA4 increased the percentage of CD8+CD69+ by 5 fold, yet this did not reduce 4T1 colony formation. This is not addressed and goes against the overall premise of the study.

Authors' response: These data (former Figure 7) have been removed and replaced with data shown in new Figure 9. The new experiment confirms that addition of anti-CD40 increases CD8+CD69+ T cells, and additionally shows that this effect of anti-CD40 is also observed when it is used with either anti-CTLA4 or RT (Figure 9D). Additional analysis of lung T cells showed that CTLA4i increased tumor-specific CD8 T cells, while anti-CD40 increased T cell activation. In all the treatment combinations tested (i.e., RT+ anti-CD40 and/or anti-CTLA4; anti-CD40 and anti-CTLA4 w/o RT) remodeling of the T cell infiltrate was associated with a significant control of lung metastases (Figure 9A). However, RT+anti-CTLA4 was already quite effective in reducing lung metastases and no further improvement was observed by the addition of anti-CD40. In contrast, addition of anti-CD40 was required to achieve control of a non-irradiated tumor in AT3 bearing mice treated with RT+CTLA4i. These results suggest that therapeutic CD40 agonism may be required for the effectiveness of RT+CTLA4i in established tumors that are dominated by an immune suppressive myeloid infiltrate such as 4T1 and AT3. However, in the settings of lung micro-metastases all combination therapies were beneficial, but none was truly superior. We have added a paragraph (page 24, lines 22-30, and 25, lines 1-2) discussing the potential clinical significance of these findings in the Discussion section.

9) Several reports are out related to the 4T1 immuno phenotype, albeit without radiation, these need to be incorporated in the discussion.

Authors' response: 4T1 is one of the most commonly used tumor models and it is not possible to cite this vast literature in an original report. We were already above the limit of references allowed and had to remove several in the revised manuscript. However, we have cited the report by Allen et al (ref 54) which describes the immune infiltrate of 4T1 and AT3 tumors and compares it to other mouse tumor models.

Minor Concern:

Although generally well written it has some a few typos / inconsistencies.

Authors' response: We have carefully read-through the manuscript and corrected all typos/discrepancies identified.

Reviewer #4

In this study, the authors studied in depth the effect of combined radiotherapy and CTLA4 inhibitor on the immune response in mice. The manuscript is well written although some parts are really dense. They combined in a first part of the study TCR sequencing, bulk and single RNAseq as well single cell TCRseq and flow-cytometry. This first part of the study is quite straightforward although I have some concerns regarding the TCR data generation quality and the analyses as well as on the single cell analysis comparison of projectTILS versus Seurat Cluster (see detailed comments).

Authors' response: We have addressed these concerns in the answer to the detailed comments.

The second part which is more flow cytometry based on a different tumor mouse model with an additional treatment added to the two first lack unfortunately a direct link with the first part to identify what the third treatment, anti-CD40, bring molecularly speaking. If this part should stay, at least RNAseq and eventually TCRseq should be added.

Authors' response: We have performed additional experiments which include RNAseq and TCRseq to identify the molecular bases of anti-CD40 effects (Figure 7 in revised manuscript).

Finally, details on the sequencing data (bulk TCR, bulk RNA and single-cell) should be added as supplemental for quality check.

Authors' response: These details have been uploaded as Supplementary Data files 1 to 7, and the content of each data file or folder is described in the end of the "Supplementary Figure and Tables" document.

Major comments:

1) The method used for bulk TCR sequencing is far from being the most reproducible as shown by previous method benchmarking (Rosati 2016, Liu 2018, Barennes 2020). ImmunoSEQ was also mentioned as being applied on some instances. When, on what? Authors should clearly specify the number of sequences obtained, the number of clonotypes at the nucleotide (nt) and amino acid (aa) level, the percentage of aligned reads.... Also, the R packages used for the analysis should be mentioned and codes published as a git if in-house developed.

Authors' response: The analysis of TCR repertoires shown in Figure 1 and associated supplementary figures was performed using iRepertoire which was available to us via our collaborator Jennifer Sims (co-author of this manuscript), who had set up this pipeline during her tenure at Memorial Sloan Kettering. One reason for choosing iRepertoire was the fact that it offered the possibility of assessing both TCR alpha and beta chains at the same time, although the chains were not paired. We are aware of the benchmarking experiment published by Barennes in 2020, and our impression is that iRepertoire generally performed well. The iRepertoire method was ranked above average based on the three criteria assessed (replicability, reliability, and sensitivity). In fact, while ImmunoSEQ scored slightly better than iRepertoire on replicability and reliability (3/10 vs 5/10), both methods scored a 3/10 for sensitivity. To assess differences between the two sequencing platforms three tumor samples sequenced using iRepertoire were also sequenced using gDNA and ImmunoSEQ (Adaptive Biotech). Although differences were found for clones with a lower representation, the repertoires were similar (new Supplementary Figure S23, also shown below).

Figure Legend: Frequency of T cell clonotypes of three tumors assessed with both Adaptive Biotech ImmunoSEQ and iRepertoire. The red dashed line indicates a diagonal line where the x-coordinate (proportion measured using the iRepertoire platform) equal the y-coordinate (proportion measured using Adaptive ImmunoSEQ platform).

All clones that were read at a proportion above approx. $1e-3$ using iRepertoire's method were also found using ImmunoSEQ. Lastly, we used the immunarch and divo packages to calculate repertoire comparisons in R. We apologies for mistakenly not mentioning this in the initial submission. This information has been added to Methods section in the revised manuscript.

2) Shannon clonality as it is calculated is not a clonality assessment, rather it seems to be the inverse of the normalized Shannon entropy, therefore not assessing clonality but a measure of the diversity, where the higher the measure the lower the diversity. This could be extrapolated as an increased clonality associated with clonal expansions but need to be demonstrated. Authors should use the Renyi profile (or Hill entropy) for instance, in order to show how much expanded clonotypes are more prevalent in the post treatment. A complementary possibility is to simply calculate the normalized cumulative frequency of the top X% (or top n sequence if the size of the repertoires are comparable). In Figure S2, the cumulative frequency of ranked rearrangements goes from 1 to 100000. Are the number of rearrangements comparable between samples? If not, a normalization is required.

Authors' response: We based our clonality calculations on the Adaptive Biotechnology's definition: "The term entropy, as applied to adaptive immune repertoires, primarily refers to the Shannon entropy, a statistic from information theory, which is one of the most robust measurements of diversity within a complex data set. Samples with higher entropy will have a greater diversity of sequences while low entropy samples will have many more sequences that share nucleotide identity. Our primary measure of entropy is calculated by summing the frequency of each clone times the log (base 2) of the same frequency over all productive reads in a sample. When this value is normalized based on the total number of productive unique sequences and subtracted from 1, a related measure, 'clonality', results."

We do appreciate that there are multiple ways to demonstrate expanded T cell clonotypes within a TCR repertoire. We chose to use calculations and measurements that have been used in the majority of scientific articles that describe TCR-sequencing data. However, we have followed the reviewer's suggestion and calculated the cumulative frequency of the top 10 clones as well as any clone with a frequency above 1%, which largely produced data similar to those shown in Figure 1. Based on the reviewer's suggestion, we also normalized our repertoires to include the top 500 clonotypes and re-calculated the above measurements. These results are now included in the revised manuscript in Supplementary Figure S3 B-E and are shown below. Overall, results are similar to the results shown in Figure 1.

3) Figure 1D shows the JSD at the VJ combination level between paired pre and post treated animals. It would be interesting to determine the similarity between the top clonotypes (including the CDR3 region) post-treatment with the whole pre-treatment repertoire in order to determine whether the post-treatment expansions result from intra-tumor expansions (including rare clonotypes that may not be captured in the top 700 from the pre-treated group) or from newly recruited cells. For this, a Jaccard and Morisita horn similarity measures might help.

Authors' response: We appreciate the suggestion from the reviewer. We did consider performing these calculations but felt that it was difficult to draw firm conclusions about expansion of specific T cells between two tissues based on descriptive TCR repertoire analysis.

4) Statistics used in the manuscript should be more detailed. In figure 1 and S3, parametric tests have been used for an n=5 max and without normality assessment. First check for normality and in all the case, given the small sample size, non-parametric test should be favored, median instead of mean should be represented and correction for multiple testing included.

Authors' response: Our apologies for not including this in the initial submission as statistical rigor is important to us. We did perform a Shapiro-Wilk normality test and it revealed that our

tumor growth data at day 21 is not significantly different from a normal distribution. As each treatment group represents its own distribution, we furthermore assessed the normality in each treatment group, and the results indicated that we can assume normality. This is shown below.

All data together: W = 0.94; p-value = 0.08
Untreated: W = 0.85376, p-value = 0.104
CTLA4: W = 0.88647, p-value = 0.2169
RT: W = 0.9675, p-value = 0.8777
RT+CTLA4: W = 0.91919, p-value = 0.4233

However, to satisfy the reviewer's curiosity regarding the non-parametric testing, we performed a non-parametric test that revealed very similar results to what is displayed in Figure 1

group1	group2	p	p.adj	p.format	p.signif	method
CTLA4	RT+CTLA4	0.00093097	0.0028	0.00093	***	Wilcoxon
CTLA4	Untreated	0.44180264	0.44	0.4418	ns	Wilcoxon
CTLA4	RT	0.0029526	0.0059	0.00295	**	Wilcoxon
RT+CTLA4	Untreated	0.00093097	0.0028	0.00093	***	Wilcoxon
RT+CTLA4	RT	0.03119977	0.037	0.0312	*	Wilcoxon
Untreated	RT	0.0062845	0.0094	0.00628	**	Wilcoxon

5) In Figure S3, when post-treatment between conditions are compared, the author should also indicate the absence of pre-treatment Shannon « clonality » between groups of mice.

Authors' response: Our apologies for not making this clearer. As seen in the former figure S3 (now figure S4), a black-filled circle without a line associated to it indicates that there is no matched sample for this specific tumor. We have clarified this in the figure legend.

6) AH1 specific-TCR signature needs to be detailed: how many unique TCRs, how related between each other, what about their distribution in the new data sets.

Authors' response: We appreciate the interest in this signature. We did not discuss this in this manuscript as the AH1-specific T cell response was thoroughly discussed and presented in our previous published report (Rudqvist et al., Cancer Immunol Res, 2018, ref. 9). In that paper, we calculate sequence similarity and present a comprehensive analysis of the AH1-specific T cell response. We have added a data file (Supplementary Data File 1) that provides the CDR3 amino acid sequence of the AH1-specific TCRs defined in ref 11 and in the present manuscript.

7) The AH1-specific TCR were inferred based on CDR3 sequences obtained from purified CD8 T-cells. Since the experiments from which they identified the AH1 specific TCRs and the new ones are done in congenic mice with the same genetic background, authors should look at the entire rearrangement (including V and J). Using previously published data on AH1 CD8 T cells is not sufficient to ensure that the CDR3s they are identifying in the tumor are from AH1 CD8 T cells.

Authors' response: We agree that inferring antigen specificity based on single beta chain alone has its limitations. However, we believe that the analysis we performed provides valuable information. We have added a sentence in discussion acknowledging the limitation of the approach (page 22, lines 12-16).

8) In Figure 2 and related text, authors should not state that the increase in CD3, CD8 and CD4 correspond to an increased infiltration as an active process, unless chemokine receptors are also observed. It could also be a proliferation due to decreased blockade of anti-inflammatory cells or molecules due to the treatment.

Authors' response: We have modified the title of figure 2 and related text in pages 6-7 and removed the word "infiltration".

8) In figure 3, authors subsampled large datasets to the lowest in order to avoid bias due to sample size. Could they detail in supplementary how many cells per condition were in fact obtained in order to appreciate the magnitude of subsampling from the larger dataset. If relevant, multiple subsamples of the largest dataset could be generated and similarity evaluate to ensure the representativeness of the subsample.

Authors' response: The original number of cells was 1,920, 2,317, 3,363, and 2,447 cells for untreated, CTLA4, RT, and RT+CTLA4 groups, respectively. This information is now provided in the method section page 33, lines 21-22.

9) Figure 3 B is surprising as it is well known that CD4 expression is usually low from single-cell data. Are the Expression levels per panel normalized per gene analyzed? If so, authors should mention it.

Authors' response: The data is normalized per gene analyzed. We filtered out cells without Cd4 or Cd8a expression, as well as double positive cells, to ensure that we were studying single positive T cells. Our scatter plot of Cd4 (y-axis) vs Cd8a (x-axis) is shown below and explains the phenomenon seen by the reviewer.

10) Then, they looked at TILs signature to deconvolute T-cell subsets and identified different cell subset. The subdivision of those ProjectTILs clusters back to the Seurat cluster is a bit more concerning. Indeed, the heatmap of the ProjectTILs signature shown in supplementary looks like there is within the Seurat clusters a high heterogeneity. Is that a biological « reality » or the results of heterogeneous gene capture? Are all the Seurat clusters consistent in terms of gene counts/UMI?

Authors' response: We agree that there is some heterogeneity within a Seurat cluster for some of the projectTILs clusters. As suggested by the reviewer we looked at total gene counts for one of the largest and most heterogeneous clusters, the Th1 projectTILs cluster (Supplementary figure 9). As shown in the figure below, unique and total gene counts do not show much difference between the different Seurat clusters. We believe that this supports the interpretation

that the heterogeneity seen within the Seurat clusters is not the result of heterogeneous gene capture.

Figure Legend: Unique and total gene count for the Seurat clusters within the Th1 population shown in Supplementary figure S9. Gene counts (y-axis) and cluster name (x-axis)

11) Finally, although I found the last part about combination of anti-CD40 together with RT and CTLA4i interesting (Figure 5 to 8), it lacks the depth and precision of the analyses shown till figure 4. Therefore, even if some link could be made between the two studies through the flow cytometry data, we cannot really what anti-CD40 is doing at the molecular level, and conclude on possible synergistic effects of the combo therapy. This part could be left for another story where some single-cell or bulk transcriptome analyses could be performed.

Authors' response: We have performed new experiments, shown in new Figures 7 and 8, to investigate the effects of anti-CD40 on the myeloid infiltrate in tumors. Briefly, we found a significant enrichment – compared to control - of a gene signature of anti-CD40 activated cDC1s only in tumors treated with RT+CTLA4i+anti-CD40 (Figure 7A and B). Consistently, flow cytometry analysis of the myeloid tumor infiltrate showed increased representation of XCR1+ cDC1 expressing higher levels of costimulatory molecules in mice treated with anti-CD40 compared to the respective treatment groups without anti-CD40. A corresponding increase in frequency and activation of XCR1+ cDC1 was also seen in the draining lymph node (Figure 8). Anti-CD40 also increased significantly AH1-specific TCR repertoire sharing between the tumor and draining lymph node of each mouse as compared to controls (Figure 7C). Overall, these data provide new mechanistic insights about the effects of anti-CD40.

Data

- It is not clear if the AH1-TCR data are from new experiments or a mixture of previous experiments (Rudqvist et al, 2018). Please clarify in the method (including the description of the signature).

Authors' response: We have added a data file (Supplementary Data File 1) that provides the CDR3 amino acid sequence of the AH1-specific TCRs defined in ref 9 and in the present manuscript.

- Newly generated TCR data should be shared via NCBI. The PRJNA596248 Bioproject does not include such data. Raw data must be obtained from iRepertoire. For TCR data, please use the template designed by the AIRR Community, which reports required metadata for AIRR-seq data sharing and reuse. If really not possible, authors could share the data through federated databases, such as iReceptor (<http://ireceptor.irmacs.sfu.ca/>) starting from tsv files. The PRJNA project indicated in the manuscript seems to target previously obtained data.

Authors' response: We attempted to upload data to iReceptor, as suggested by the reviewer, but were informed that *“it is not possible to upload TCR sequences to the iReceptor Gateway. The iReceptor Gateway is a data discovery and analysis platform that queries the AIRR Data Commons, but it does not provide a mechanism to upload your own data”*. Thus, to share our data openly, we have included all TCR repertoire data tables and annotated sample information data spreadsheet as additional file archive associated to this manuscript (Supplementary Data Files 5, 6 and 7).

The PRJNA596248 Bioproject indeed includes DNA and RNA sequencing data associated with a published manuscript – in addition to the data associated with the current study. To avoid confusion we have added an excel file (Supplementary data file 4) that lists the data related to current manuscript Figures 2 and 7.

Minor comments

Several typos to be corrected across the manuscript

Authors' response: We have carefully read-through the manuscript and corrected all typos/discrepancies identified.

REVIEWERS' COMMENTS

Reviewer #1 (Remarks to the Author):

The authors has addressed the majority of reviewer concerns and additional experiments and data have significantly improved this manuscript.

One minor comment is the abscopal effect in 4T1. I would suggest presenting the response in 4T1 as preventing metastatic seeding rather than calling it an abscopal response as typically an abscopal response refers to shrinkage of measurable disease.

Reviewer #2 (Remarks to the Author):

The authors have addressed my concerns.

Reviewer #3 (Remarks to the Author):

The authors addressed all my previously raised concerns.

Reviewer #4 (Remarks to the Author):

The authors have mainly addressed most of my concerns.

RE: NCOMMS-22-43189

Immunotherapy targeting different immune compartments in combination with radiation therapy induces regression of resistant tumors

Reviewer #1

The authors has addressed the majority of reviewer concerns and additional experiments and data have significantly improved this manuscript.

One minor comment is the abscopal effect in 4T1. I would suggest presenting the response in 4T1 as preventing metastatic seeding rather than calling it an abscopal response as typically an abscopal response refers to shrinkage of measurable disease.

Authors' response: We agree with the reviewer. In the manuscript we do not refer to the control of lung metastases in the 4T1 model as abscopal effect.